# Synergistic binding sites in a metal-organic framework for the optical sensing of nitrogen dioxide

Isabel del Castillo-Velilla [1,8], Ahmad Sousaraei[1,8], Ignacio Romero-Muñiz [1], Celia Castillo-Blas[1], Alba S. J. Méndez [2], Freddy E. Oropeza [3], Víctor A. de la Peña O'Shea [3], Juan Cabanillas-González [4], Andreas Mavrandonakis [5] ✉ & Ana E. Platero-Prats [1,6,7] ✉

Luminescent metal-organic frameworks are an emerging class of optical sensors, able to capture and detect toxic gases. Herein, we report the incorporation of synergistic binding sites in MOF-808 through post-synthetic modification with copper for optical sensing of $NO_2$ at remarkably low concentrations. Computational modelling and advanced synchrotron characterization tools are applied to elucidate the atomic structure of the copper sites. The excellent performance of Cu-MOF-808 is explained by the synergistic effect between the hydroxo/aquo-terminated $Zr_6O_8$ clusters and the copper-hydroxo single sites, where $NO_2$ is adsorbed through combined dispersive- and metal-bonding interactions.

Mononitrogen oxides ($NO_x$) are an important class of hazardous gases that are produced in the air during high-temperature combustion as a result of the reaction of $N_2$ with $O_2$. $NO_2$ sensors are typically based on chemical reactions that can generate either light or/and small electric current.[1–3] Traditional sensors operate via indirect measurements, thereby requiring both accurate calibration as well as minimisation of interferences such as water vapour and reactive nitrogen compounds. Therefore, the development of advanced materials capable of detecting $NO_2$, directly, with high sensitivity at room temperature, is pivotal to target the next generation of sensing technologies.

Metal–organic frameworks (MOFs) are porous scaffolds constructed through linking inorganic nodes by organic linkers in an orchestrated manner[4]. These materials possess highly desirable properties such as permanent porosity, tunable architectures, or luminescent properties[5], among others. Specifically, luminescent MOFs have been extensively exploited in chemical sensing to detect a wide range of toxic gases[6–8]. Some of us have reported a family of

lanthanide MOFs containing linkers with amino groups that act as recognition centres for $NO_2$[3]. The interaction between nitrogen dioxide ($NO_2$) and amino groups has been observed to elicit a notable response in the luminescence of lanthanide-based sensors. Surprisingly, the nature of this response is contingent upon the particular lanthanide utilised, with either an increase or decrease in luminescence being reversible in nature. This novel sensing scheme demonstrates significant potential for photoluminescent detection. Therefore, luminescent MOF materials are promising platforms for high-performance $NO_2$ sensing technologies. Defect engineering in MOFs has acquired remarkable attention in the field of gas adsorption[9]. Zr-MOFs may have structural defects without compromising their stability[10]. One strategy to incorporate functional defects in robust Zr-MOFs consists of modifying the inorganic $Zr_6O_8$ clusters with metal cations[11–17]. In particular, the incorporation of copper in Zr-MOFs can afford either the stabilisation of Cu(II)[15,16] or Cu(I)[17] as single sites. Within MOF-808, the incorporation of copper can be done

[1]Departamento de Química Inorgánica, Facultad de Ciencias, Universidad Autónoma de Madrid, 28049 Madrid, Spain. [2]Deutsches Elektronen-Synchrotron DESY, Notkestraße 85, 22607 Hamburg, Germany. [3]Photoactivated Processes Unit, IMDEA Energy, Parque Tecnológico de Móstoles, Avenida Ramón de la Sagra 3, 28935 Móstoles, Madrid, Spain. [4]Madrid Institute for Advanced Studies, IMDEA Nanociencia, c/ Faraday 9, Campus de Cantoblanco 28049 Madrid, Spain. [5]Electrochemical Processes Unit, IMDEA Energy, Parque Tecnológico de Móstoles, Avda. Ramón de la Sagra 3, 28935 Móstoles, Spain. [6]Condensed Matter Physics Center (IFIMAC), Universidad Autónoma de Madrid, 28049 Madrid, Spain. [7]Institute for Advanced Research in Chemical Sciences (IAdChem), Universidad Autónoma de Madrid, 28049 Madrid, Spain. [8]These authors contributed equally: Isabel del Castillo-Velilla, Ahmad Sousaraei. ✉ e-mail: andreas.mavrantonakis@imdea.org; ana.platero@uam.es

through either the insertion of functional groups[18,19] or the metalation of reactive aqua ligands within the $Zr_6O_8$ clusters[20]. This MOF is composed of hydroxo/aquo-terminated $Zr_6O_8$ clusters linked by benzene-1,3,5-tricarboxilate (BTC) ligands (Fig. 1)[21]. Recently, the adsorption of $NO_2$ has been demonstrated for a series of functionalized MOF-808 analogues[22]. Despite these promising results, the partial irreversible nature of $NO_2$ adsorption on MOF-808, together with slow kinetics, were identified. In this context, we envisaged that the modification of MOF-808 with metal sites able to bind $NO_2$ may expand the potential sensing applications of this material.

Here we show the luminescent performance of MOF-808 containing copper single sites as a promising optical sensor to monitor low concentrations of $NO_2$. Modelling tools have been applied to better understand the molecular interactions and the binding mechanism between $NO_2$ and this MOF. To gain insight into the atomic structure of copper sites capable of reversibly bind to $NO_2$, a comprehensive analysis was undertaken, combining advanced synchrotron characterisation techniques such as pair distribution function (PDF) analysis of X-ray total scattering data and Cu and Zr K-edge X-ray absorption spectroscopy (XAS). Additionally, density functional theory (DFT) calculations were employed to expand the experimental findings. The results of this study provide a deeper understanding of the binding mechanism of $NO_2$ to the copper sites in Cu-MOF-808 under investigation.

## Results and discussion

### Post-synthesis modification

MOF-808 was synthesised and activated by treatment with 1 M HCl (see Methods, Supplementary Note 1 and Supplementary Fig. 1). Metalation of MOF-808 was carried out by treatment with a solution of MeOH containing copper(II) acetate monohydrate (see Supplementary Note 1 and Supplementary Fig. 2). Structure retaining and long-range order of the Cu-MOF-808 material was corroborated by powder X-ray diffraction data (PXRD) (Fig. 2D). Chemical analyses of Cu-MOF-808 indicated the presence of ca. 3.3 Cu atoms per $Zr_6O_8$ cluster. Scanning electron microscopy and energy dispersive X-ray spectroscopy (SEM-EDX) showed the homogenous incorporation of copper within MOF-808

(Fig. 2C and Supplementary Note 4). Nitrogen adsorption-desorption isotherms collected on the Cu-MOF-808 indicated an expected decrease in specific surface area after copper metalation (from 1063 to 520 $m^2 g^{-1}$ for activated MOF-808 and Cu-MOF-808, respectively), with a significant decrease in volume linked to the mesopores of 18 Å (ca. 60%) (Fig. 2A, B and Supplementary Note 5).

### X-ray absorption spectroscopy

Variable temperature XAS experiments were performed to elucidate both the geometry and the oxidation state of copper within Cu-MOF-808. The X-ray absorption near-edge structure (XANES) data obtained at 298 K show two main features (Fig. 3). The pre-edge peak linked to the quadrupole-allowed $1s{\rightarrow}3d$ transition, is seen at 8979 eV, in agreement with the presence of Cu(II) in a twisted-square-planar geometry[23]. An additional feature is observed at 8986 eV, which is associated with the $1s{\rightarrow}\{4p + shakedown\}$ transition. Derivative analyses of the XANES data collected at 10 K showed a strong temperature dependence of the signal intensity in this region (Fig. 3B), which corroborates the mixing between $3d$ and $4p$ metal orbitals[24]. EXAFS data showed a predominant contribution at ca. 1.97 Å that corresponds to Cu(II)-O bonds.

### Pair distribution function analyses

PDF analyses of X-ray total scattering data were performed to assess the atomic structure of the copper sites stabilised into the MOF-808 framework. Differential analysis of the PDF data (dPDF) was carried out to identify the correlations associated with the copper sites within the MOF. In particular, dPDF analysis of Cu-MOF-808 showed bond contributions attributed to Cu(II)−O distance and Cu⋯M interactions at 2.06 and 3.34 Å, respectively (Fig. 4). According to both the EXAFS data and the differences of X-ray scattering power of Cu and Zr, the most plausible assignment is Cu⋯Zr[25]. We hypothesised the presence of copper single sites attached to MOF-808 through hydroxo or aquo defect sites within the $Zr_6O_8$ cluster.

### Structural modelling

Density functional theory (DFT) calculations were performed in order to elucidate the precise configurations of Cu-MOF-808. The structural

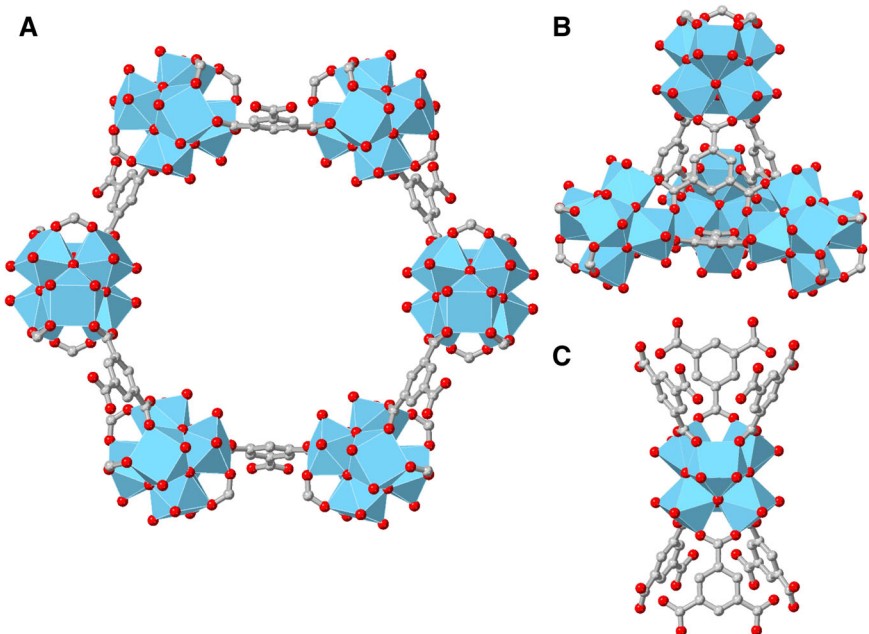

**Fig. 1 | Structural representation of MOF-808 showing the presence of hexagonal and tetrahedral cavities. A** Representation of a hexagonal pore and **B** a tetrahedral cavity in MOF-808. **C** Depiction of an unsaturated $Zr_6O_8$ cluster in MOF-808. Colour scheme: blue = Zr, grey = C, red = O; hydrogen atoms have been omitted for clarity.

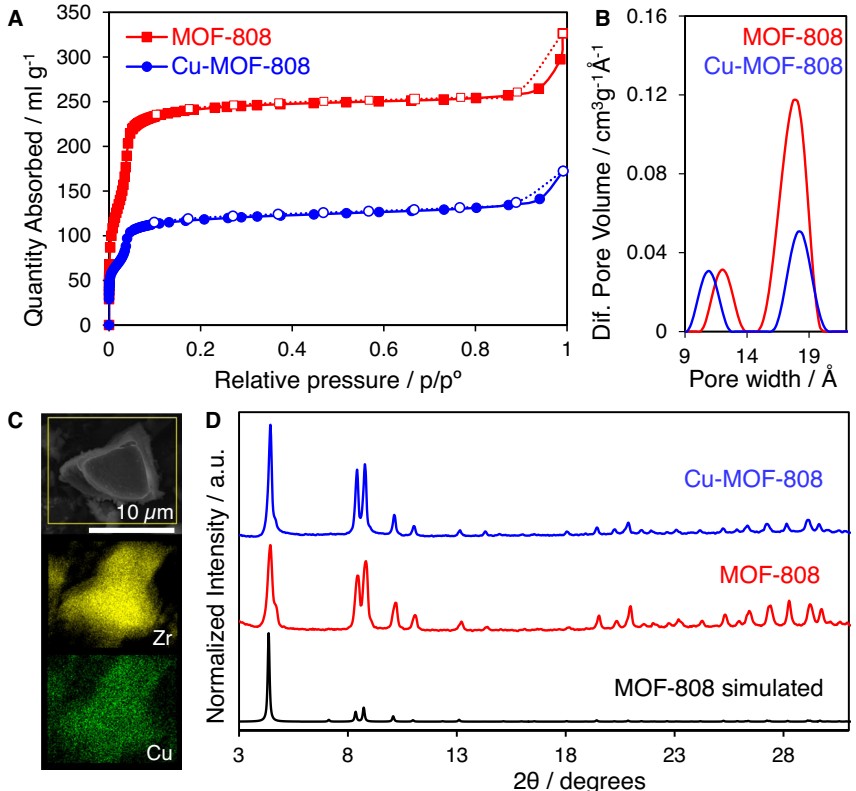

**Fig. 2 | Structural and textural characterisation of Cu-MOF-808. A** $N_2$ adsorption/desorption isotherms at 77 K demonstrate a reduction in the surface area following copper incorporation. **B** Pore size distribution (PSD) analysis shows differences between pristine and Cu-MOF-808 material. **C** Elemental mapping highlights the spatial distribution of Zr (yellow) and Cu (green) in Cu-MOF-808. **D** PXRD data of the synthesised materials compared with calculated data for the MOF-808 crystal structure.

and energetic properties of several mono- and bi-nuclear copper-oxo and copper-hydroxo clusters deposited on the nodes of the MOF-808 were computed (see Supporting Information for a detailed explanation of the methodology, the computed models and the definition of formation energies). Based on our previous work on Fe-MOF-808[20], several possible models have been considered for the deposition of two copper atoms between neighbouring $Zr_6O_8$ nodes. Here, the three most stable configurations are presented, for which the most negative formation energies have been calculated. Similarly to the Fe-MOF-808 behaviour, the deposition as two isolated single metal sites connected with the $\mu_3$- and terminal oxygen atoms of the nodes (model A-2Cu) is energetically favoured by ~41 (-35) kJ/mol with respect to two single metal sites connected with the terminal oxygen atoms of the nodes (model B-2Cu) according to the r2-SCAN-3c (M06L). Several other structures have been considered, where the two copper atoms are linked to the $Zr_6O_8$ nodes via terminal oxygen atoms, and are bridged by $\mu_2$-O, $\mu_2$-OH, or $\mu_2$-OH$_2$. Among them, the most stable configuration is with a $\mu_2$-OH bridge (model C-2Cu), and is ~35 kJ/mol (14 kJ/mol) higher than model A-2Cu according to the r2-SCAN-3c (M06L), shown in Fig. 5A. All models are presented in the Supplementary Note 12 (Supplementary Table 4 and Supplementary Fig. 24). The characteristics, i.e. the distorted planar coordination environment of the copper atom, and the Cu⋯Cu and Cu⋯Zr distances, for models B and C indicate very good agreement with the experimental PDF data. While model A is lower in energy, it is not a good match based on the above-mentioned geometrical characteristics. We think that models B and C are matching better the experimental structures, although they are predicted to be slightly higher in energy. One possible explanation for the experimental observation of B and C, is the higher barrier for the deprotonation of the $\mu_3$-OH by the copper precursor, and thus the

kinetics are much slower for the formation of model A. Therefore, our calculations favour the hypothesis of copper single sites between the $Zr_6O_8$ nodes and along the edges of the tetrahedral cavities in MOF-808, resulting in a unique hetero-bimetallic structure (Fig. 5B, C).

**Optical sensing**

The NO$_2$ sensing capacity of Cu-MOF-808 was studied upon exposure to 50 ppm NO$_2$ in comparison with MOF-808. Figure 6 illustrates the NO$_2$ sensing measurements collected for Cu-MOF-808 and MOF-808 at room temperature. The photoluminescence (PL) spectra on both MOFs were obtained in real-time during four N$_2$-NO$_2$ cycles, (16 min per cycle; 87 s between each spectral acquisition). According to PL kinetics, Cu-MOF showed significant PL quenching (≤45%) in the presence of NO$_2$ and reversibility through N$_2$ purging (Fig. 6A, blue squares). On the other hand, MOF-808 exhibited a drastically lower response to the N$_2$-NO$_2$ cycles, along with a decreasing trend for its performance (Fig. 6A black triangles). Remarkably, Cu-MOF-808 PL changes are still observed when the NO$_2$ concentration is reduced to 25 and 10 ppm (Fig. 6B). As shown in Fig. 6C, Cu-MOF-808 has a good sensing response even after 2 h of continuous exposure to 50 ppm of NO$_2$, whilst MOF-808 no longer respond to NO$_2$ gas. The materials were then exposed to different concentrations of NO$_2$ to assess the limit of detection (LOD), resulting in values of 101 and 16 ppb for MOF-808 and Cu-MOF-808, respectively (see Supplementary Fig. 32). Furthermore, the sensitivity of the Cu-MOF-808 to other common gases present in air potentially able to bind to the copper sites were explored, such as CO$_2$ and O$_2$. The results confirmed a significantly higher sensitivity of Cu-MOF-808 to NO$_2$ compared to CO$_2$ and O$_2$ (see Supplementary Fig. 33). These results demonstrate the best optical response reported so far for a MOF material towards the sensing of NO$_2$.

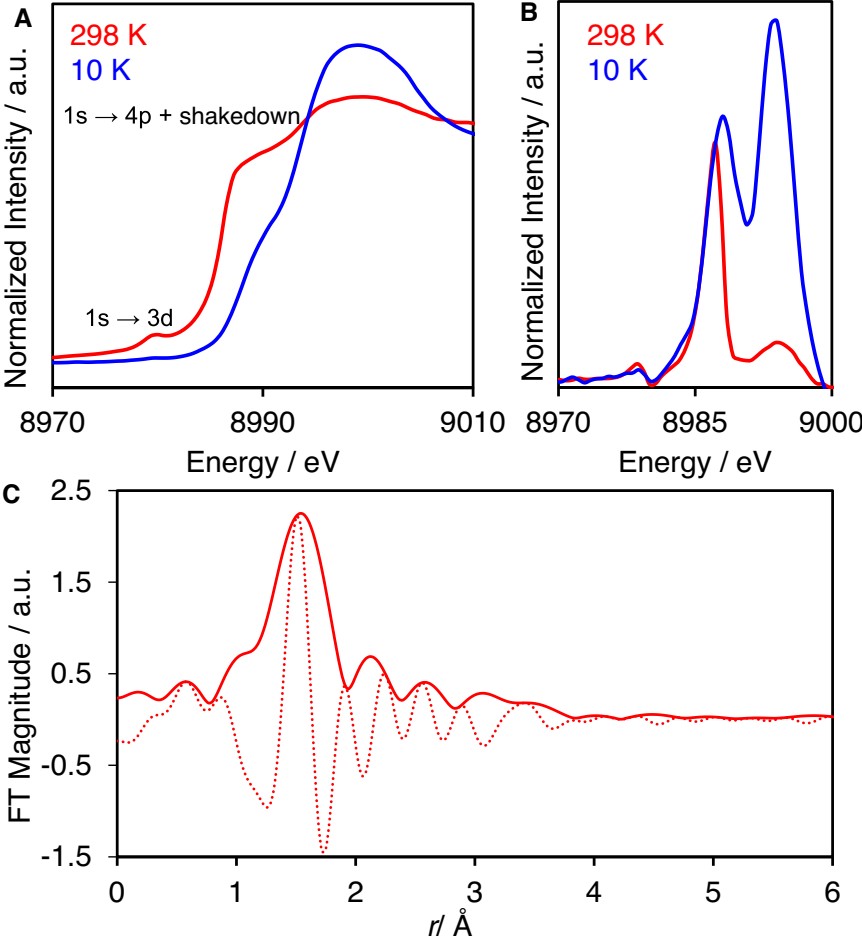

**Fig. 3 | Variable temperature X-ray absorption spectroscopy data of Cu-MOF-808. A** Cu *K*-edge XANES and **B** First derivative spectra for Cu-MOF-808 collected at 10 and 298 K. **C** k$^2$-weighted Cu *K*-edge Mag[χ(R)] (solid line) and Im[χ(R)] (dash line) spectra of Cu-MOF-808 at 10 K.

To shed light into the sensing mechanism, we first measured the PXRD to understand whether the NO$_2$ inclusion interfered with the Cu-MOF-808 crystallinity. Figure 6D shows the PXRD data of Cu-MOF-808 before and after NO$_2$ sensing, demonstrating the retaining of the MOF structure. FT-IR spectroscopy data collected on Cu-MOF-808 before and after the sensing experiments revealed the absence of major changes (see Supplementary Fig. 14), showing the stability of the framework after exposure to NO$_2$. To gain insight into the local nature of the adsorption of NO$_2$ within Cu-MOF-808, X-ray photoelectron spectroscopy (XPS) experiments were performed to elucidate the binding of NO$_2$ to the copper centres. As shown in Supplementary Fig. 22, XPS spectra of Cu-MOF-808 before and after exposure to NO$_2$ present feature peaks in the Cu 2*p* region at 934.1 eV value well within the expected binding energy for Cu(II) cations. The Cu 2*p* region also includes high binding energy satellites characteristic of Cu(II) in a *d*$^9$ electron configuration. This fact evidences the lack of variations in terms of the oxidation state of the copper sites under exposure to NO$_2$. Interestingly, the N1*s* region of the NO$_2$-loaded Cu-MOF-808 Supplementary Fig. 21 can be fitted with two components centred at binding energies 407.1 and 404.4 eV. The high binding energy component matches closely the species of NO$_2$ adsorbed on metal-oxide surfaces[26, 27], and therefore we tentatively assigned this component to species formed upon NO$_2$ adsorption on the Zr$_6$O$_8$ centres of the MOF sample. On the other hand, the low binding energy component centred at 404.4 eV could be assigned to partially reduced NO$_2$ species interacting with the copper-oxo sites, similar to those occurring upon NO$_2$ adsorption on Cu$_2$O (111) surfaces[28]. Based on DFT calculations, in the interaction of Cu centres with NO$_2$, there is a net transfer of electronic charge to the NO$_2$ molecule, which leads to a partial reduction, consistent with the observation of a low binding energy component in the N1*s* region.

## Mechanism of NO$_2$ binding

As a next step, the interaction energies of NO$_2$ with the models A, B and C were calculated. In all cases, the nitrogen dioxide is bound via the nitrogen atom to the copper atom (Fig. 7). The corresponding calculated binding energies for models A and B are −64.3 and −38.3 kJ/mol, respectively. In model C, two configurations leading to two different binding energies (−59.1 and −48.5 kJ/mol) were considered, because the two copper atoms have different environments. In the most stable adsorption geometries in models A and C, the NO$_2$ is further stabilised through additional dispersion interactions with the μ$_3$-OH.

Additional calculations were also performed in order to estimate the individual contributions of the NO$_2$ binding due to the dispersive- and metal-bonding interactions. For the first case, the interaction energies of NO$_2$ with the pristine MOF-808 were calculated. Several initial configurations have been considered and the interaction energies of NO$_2$ with MOF-808 are computed between approximately −22 and −29 kJ/mol. The most stable adsorption geometry is shown in Fig. 7 and Supplementary Fig. 25. For the latter case, the HKUST-1 MOF has been chosen as the model system to study the Cu(II)-NO$_2$ interactions, because it contains coordinatively unsaturated copper atoms in a distorted square planar environment. Detailed results can be found in Supplementary Note 12 (Supplementary Table 5 and Supplementary

Fig. 25). By using the reported structures from previous work, the interaction energies of $NO_2$ with the copper atom of HKUST–1 are computed at approximately −32 and −28 kJ/mol for binding via the oxygen and nitrogen atoms respectively[29]. The sum of these two individual contributions is in the range of −52 till −59 kJ/mol, which agrees very well with the value of −59.1 kJ/mol calculated for the binding of $NO_2$ with the model C-2Cu. Noteworthy, the adsorption geometry of $NO_2$ is different in the Cu-MOF-808 than in the HKUST-1. While $NO_2$ adsorption in the HKUST-1 occurs through the oxygen

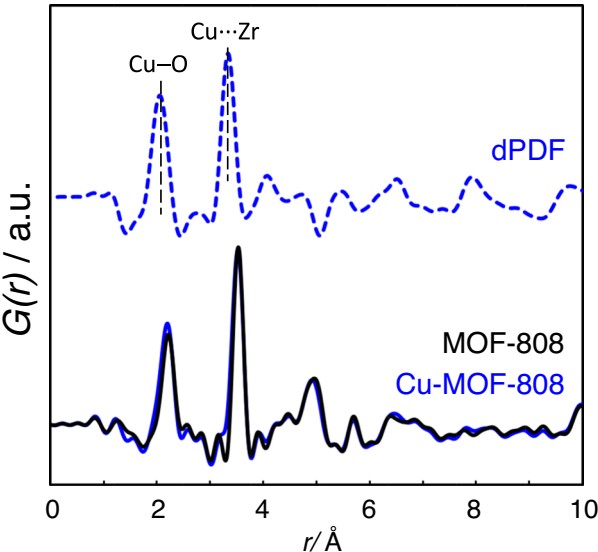

**Fig. 4 | Local structural analyses of Cu-MOF-808 by applying differential pair distribution function analysis.** Differential analysis of the PDF data (dPDF) of Cu-MOF-808 (up) was obtained after subtracting the total PDF data of pristine MOF-808 to that of Cu-MOF-808 (down).

atom, the presence of additional dispersive bonding in the Cu-MOF-808 makes the adsorption via the nitrogen atom more favourable.

The different nature of the interactions of $NO_2$ with MOF-808 and Cu-MOF-808 is also demonstrated by performing an Interaction Region Indicator (IRI) analysis, which is able to reveal simultaneously weak and strong chemical-bonding interactions[30]. The analysis is done for the optimised geometry of $NO_2$ with model C-2Cu and with the MOF-808 model (see Fig. 7 and Supplementary Fig. 25). The colouring scheme and its translation into interactions is shown in Supplementary Note 12 (Supplementary Fig. 26). It can be seen that in the case of the parent MOF-808, the interactions with $NO_2$ are dominated by weak dispersive forces (vdW interactions shown as isosurfaces of green colour). In the case of Cu-MOF-808, the interactions are attributed to both metal-bonding and dispersive nature, as revealed by the presence of isosurfaces in dark blue and green colour, respectively. Moreover, the difference in the nature of interactions of $NO_2$ with Cu-MOF-808 and MOF-808 are also depicted in the significant charge transfer of 0.23|e| towards the LUMO orbitals of $NO_2$ upon interaction with the copper sites. A charge population analysis based on AIM and also CM5 charge schemes indicates that the copper partial charge remains unchanged before and after $NO_2$ binding. The analysis reveals a complex mechanism, where a charge is transferred from the MOF framework to the $NO_2$ through the copper sites. The charge is shifted mainly from the carbon atoms of the carboxylate groups. This charge depletion from the framework atoms can explain the luminescence quenching of the Cu-MOF-808 in the presence of $NO_2$. When $NO_2$ interacts with the parent MOF-808, no charge transfer is computed. Therefore, the adsorption mechanism of $NO_2$ in the Cu-MOF-808 can be explained as a combination of weak dispersive- and stronger metal-bonding interactions.

We have also explored computationally the binding of competing gases (such as $H_2O$, NO, CO, and $CO_2$) with the Cu-MOF-808. The calculations predict that the competing gases are adsorbed less strongly on the copper sites than the $NO_2$. In the case of $H_2O$, the preferred adsorption position is not on the copper site, but close to the $ZrO_2$ node, where the water molecule can interact via hydrogen bonds with

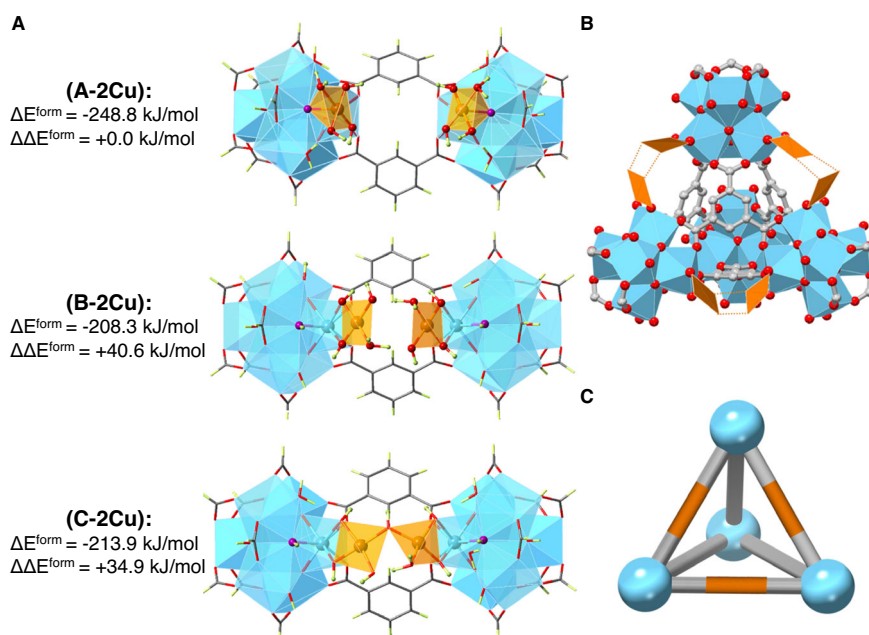

**Fig. 5 | Models of copper-hydroxide species deposited on the MOF-808 nodes. A** Formation energies ($\Delta E^{form}$) and relative formation energies ($\Delta\Delta E^{form}$) in kJ/mol for the deposition of two copper-hydroxide species on the MOF-808 nodes. The three most stable configurations are presented here. The formation energies are calculated by considering a reaction of the MOF-808 with two copper precursor species and the subsequent release of water molecules. The relative formation energies are referred to versus the formation energy of the model A-2Cu. **B** Structural and **C** schematic representation of the hetero-bimetallic tetrahedral structural subunit in Cu-MOF-808. Colour scheme: blue = Zr, grey = C, magenta = $\mu_3$-O, orange = Cu, red = O, and green = H.

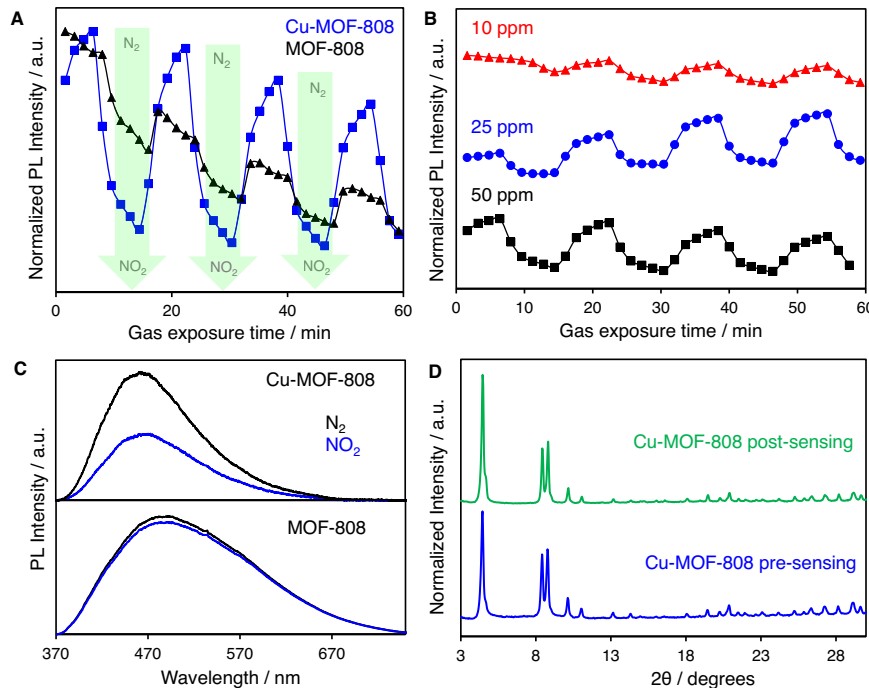

**Fig. 6 | Performance of Cu-MOF-808 as an optical sensor for the detection of NO₂ and stability studies. A** Photoluminescence (PL) kinetics of Cu-MOF-808 and MOF-808 under 50 ppm of NO₂, measuring at 473 and 480 nm, respectively, **B** Sensing response of Cu-MOF-808 at 50 (black- squares), 25 (blue- circles), and 10 ppm (red- triangles) concentrations, **C** Sensing performance of Cu-MOF-808 and MOF-808 after 2 h sensing measurements, ($_{exc}$ = 355 nm), **D** PXRD analyses of Cu-MOF-808 before and after the NO₂ sensing measurement. (The N₂-NO₂ exposition is shown by the shaded areas).

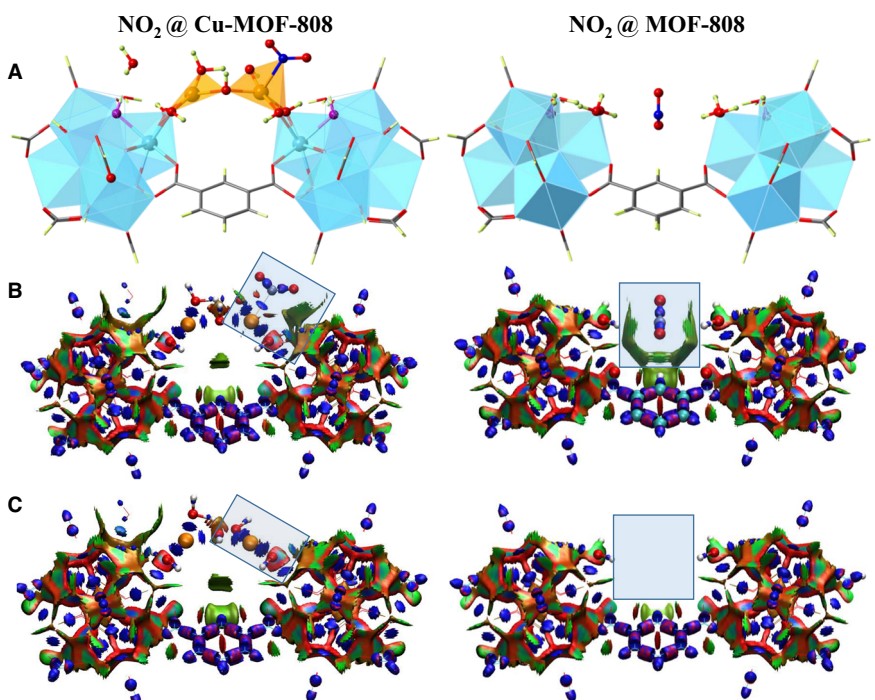

**Fig. 7 | Interaction of NO₂ with the Cu-MOF-808 and MOF-808. A** Adsorption geometries of NO₂ in Cu-MOF-808 (left) and MOF-808 (right). Colour scheme: blue = Zr, grey = C, magenta = $\mu_3$-O, orange = Cu, red = O, green = H. **B** Interaction region indicator (IRI) plots for NO₂ interacting with Cu-MOF-808 (left) and MOF-808 (right). **C** IRI plots for the Cu-MOF-808 (left) and MOF-808 (right). The adsorption region is highlighted with a rectangular box. The colouring scheme is also explained in Supplementary Note 12 (Supplementary Fig. 26).

the hydroxo and aqua ligands of the node. Detailed results can be found in Supplementary Note 12 (Supplementary Table 6).

In this work, we successfully incorporate synergistic binding sites in MOF-808, resulting in excellent optical sensing properties towards

NO₂ at remarkably low concentrations (LOD = 16 ppb). Computational modelling and advanced synchrotron characterisation tools such as PDF and XAS are combined to elucidate the precise atomic structure of Cu-MOF-808. In particular, we demonstrate that copper-hydroxo

single sites are deposited between neighbouring $Zr_6O_8$ nodes in the MOF-808 framework, resulting in a unique hetero-bimetallic tetrahedral structural subunit. DFT calculations have shed light on the interaction mechanism of $NO_2$ with Cu-MOF-808, linked to the synergistic occurrence of weak dispersive- and stronger metal-bonding interactions between $NO_2$ and the $Zr_6O_8$ clusters post-synthetically modified with copper-hydroxo single sites. This work opens an exciting horizon in the field of optical sensing of toxic gases using MOF materials by harnessing the occurrence of weak-binding interactions through post-synthetic chemical modifications. Further studies on the use of this material to detect a variety of environmental pollutants with reversible mechanisms are currently underway.

## Methods

### Synthesis
All reagents were used as received from commercial suppliers unless otherwise stated.

### MOF-808
Trimesic acid (210 mg and 1.0 mmol) and $ZrOCl_2 \cdot 8H_2O$ (970 mg and 3.0 mmol) were added to a mixture of 90 ml of formic acid (45 mL) and DMF (45 mL) in a screw cup glass bottle. The reaction was heated at 130 °C for 48 h in the oven. After cooling to room temperature, white powder was collected by centrifugation (13,680×$g$, 2 min), and the solid was washed with DMF, distilled water and acetone (50 mL × 3 each). The solid was dried in the oven at 60 °C overnight. Then, the product (800 mg) was put in a solution of 1 M of HCl (100 ml), stirring at room temperature for 24 h, with the aim of exchanging some of the formates ligands by water. The resulting mixture was centrifuged and the solid was washed with distilled water and acetone (50 mL × 3). The solid was dried in the oven at 60 °C overnight, yielding MOF-808 as a white powder (750 mg).

### Cu-MOF-808
A mixture of MOF-808 treated with HCl (100 mg and 0.056 mmol) and $Cu(CH_3COO)_2 \cdot H_2O$ (68 mg and 0.34 mmol) and MeOH (10 mL) was placed in a sealed vial. The reaction was stirred at 60 °C overnight. After cooling to room temperature, the reaction mixture was centrifuged, and the solid was washed with MeOH, distilled water and acetone (three times with 10 mL each). The solid was dried in the oven at 60 °C overnight, yielding Cu-MOF-808 as a turquoise powder (85 mg). Chemical formula: $[Zr_6Cu_{3.3}O_8H_4(C_9H_3O_6)_2(OH)_{9.1}(H_2O)_{3.2}(C_2H_3O_2)_{3.5}] \cdot (C_3H_7ON)_{0.4}$.

### Powder X-ray diffraction
Powder X-ray diffraction (PXRD) measurements were performed using a Bruker D8 diffractometer equipped with a copper source operating at 1600 W. The samples were ground and placed onto a borosilicate sample holder, and the surface was levelled with a clean microscope slide. The diffraction patterns were collected in continuous mode over a 2θ range of 3 to 45 degrees, with a step size of 0.02° and an exposure time of 0.5 s per step.

### X-ray total scattering
Synchrotron X-ray total scattering data suitable for pair distribution function (PDF) analyses were acquired at the P02.1 beamline at PETRA III (Deutsches Elektronen-Synchrotron) using 60 keV (0.207 Å) X-rays. Samples were loaded into 0.8 mm diameter polyamide (kapton) capillaries and sealed with epoxy. The data were collected using a Varex XRD 4343CT area detector equipped with a CsI scintillator directly deposited on amorphous Si photodiodes. The detector had a $150 \times 150 \, \mu m^2$ pixel size and a 2880 × 2880 pixel area. Geometric corrections and data reduction to the 1D format were performed using DAWN Science software[31]. To generate PDFs, the PDFgetX3 programme[32] within the xPDFsuite software package was utilised, with

a $Q_{max}$ value of 22 Å$^{-1}$. Differential analyses were performed by subtracting the total PDF data of Cu-MOF-808 from that of pristine MOF-808 in real space, following normalisation.

### X-ray photoelectron spectroscopy
X-ray photoelectron spectra were collected using a lab-based spectrometer (SPECS GmbH, Berlin) equipped with a monochromated Al source (Al Kα$_1$ hν = 1486.6 eV) operated at 50 W. The X-ray was focused onto the sample with a μ-FOCUS 600 monochromator, with a spot size of 300 μm. The data was recorded in fixed analyser transmission (FAT) mode using a PHOIBOS 150 NAP 1D-DLD analyser. The pass energy was set to 40 eV for survey scans and 20 eV for high-resolution regions. To calibrate the binding energy scale, the Au $4f_{7/2}$ (84.01 eV) and Ag $3d_{5/2}$ (368.20 eV) peaks were used. Charge compensation was required during data acquisition, and recorded spectra were calibrated against the C1$s$ internal reference. Data interpretation was performed using Casa XPS software, with Shirley or two-point linear background used depending on the spectrum shape. Surface chemical analysis was performed based on the peak areas of the high-resolution spectra, with the CasaXPS sensitivity factors used for quantification (where the relative sensitivity factor of C1s is 1.000).

### Scanning electron microscopy
Scanning electron microscopy images were collected with a JEOL JSM 7600 F microscope. Energy dispersive X-ray spectra (EDS) were collected with an S-3000N microscope equipped with an ESED and an INCAx sight of Oxford Instruments. All samples were prepared by dispersing the material onto a double-sided adhesive conductive carbon tape that was attached to a flat aluminium sample holder and they were sputtered with carbon or gold (12 nm).

### Nuclear magnetic resonance
Spectra were acquired on a Bruker AV-300 spectrometer, running at 300 MHz for $^1H$. Chemical shifts (δ) are reported in ppm relative to the residual solvent signal with a value of 2.50 ppm for DMSO-d$_6$. $^1H$ digested solution NMR (100 μL D$_2$O, 1 mg NaF, 50 μL HF and 500 μL DMSO-d$_6$) of as-synthesised sample MOF-808.

### Textural analyses
The Micromeritics ASAP 2020 system was used to measure nitrogen adsorption and desorption isotherms at 77 K after outgassing the samples at 100 °C for 16 h. The specific surface area (BET) was determined using the Brunauer–Emmett–Teller equation, with the nitrogen molecule area taken as 0.162 nm². While the linear range of the BET equation was between 0.05–0.35 P/P$_0$ for some materials, it was much narrower and displaced to lower relative pressures (P/P$_0$ = 0.04–0.07) for the microporous materials studied. Micropore volume and external surface area were determined using t-plot analysis based on the assumption that the thickness of an adsorbed layer of nitrogen was 0.354 nm, and its arrangement was hexagonal close-packed. Mesopore volumes were calculated by subtracting the microporosity from the volume of gas adsorbed at a relative pressure of 0.6 on the desorption branch of the isotherms, which corresponds to the filling of all pores below 50 nm. The total pore volume was determined by the volume of gas adsorbed at a relative pressure of 0.95 on the adsorption branch of the isotherms. Pore size distribution (PSD) curves were obtained from the adsorption branches using the non-local density functional theory (NLDFT) method for a cylinder pore in pillared clays, using a regularisation of 0.100. MicroActive software was used for these analyses.

### Thermogravimetric and differential thermal analyses
Data were collected using an SDT Q600 from TA Instrument equipment in a temperature range between 20 and 800 °C in an air atmosphere (100 mL/min flow) and a heating rate of 10 °C/min.

## Fourier-transform infrared spectroscopy

Spectra were recorded on a PerkinElmer 100 spectrophotometer using a PIKE Technologies MIRacle Single Reflection Horizontal ATR Accessory from 4000–450 cm$^{-1}$.

## Compositional analyses

Elemental analyses were performed with a LECO CHNS-932 analyser. Induced coupled plasma emission spectroscopy was performed with an ICP PerkinElmer mod. OPTIMA 2100 DV equipment. Samples (3 mg) were digested in 4 mL of a 1:1 $H_2O_2$:$H_2SO_4$ mixture (v:v) and taken to 10 mL in a volumetric flask volume with distilled water.

## Computational details

Density functional theory (DFT) calculations were performed in order to elucidate the possible configurations of Cu-MOF-808. See Supplementary Note 11 for details about the computational methodology. We modelled the structural and energetic properties of several mono- and bi-nuclear copper-oxo and copper-hydroxo clusters deposited on the nodes of the MOF-808. As a starting model for the pristine MOF-808, we choose a molecular cluster that is composed of two $Zr_6O_8$ octahedra bridged by two ligands. This model has been previously used in our previous work, where the deposition of iron-oxo clusters on the MOF-808 was investigated[20]. The coordinates of the starting model are carved from the experimentally determined crystal structure. The benzene-tricarboxylate ligands, which are bridging the two $Zr_6O_8$ octahedra, are cropped to benzene-dicarboxylate. The remaining four ligands are also cropped to formate. Based on the experimental observations, six (6) formate molecules are further added as capping ligands. For charge balancing, four (4) protons have to be added to the $\mu_3$-O atoms of each $Zr_6O_8$ octahedron. As a next step, four formate capping ligands are removed, with each one being replaced by a hydroxo and a water molecule, giving rise to the MOF-808 model, where the copper-hydroxide species will be deposited.

Subsequently, we investigated the structural and energetic characterisics of the deposition of two Cu(II) atoms on the nodes of the MOF-808 model. Two possible ways of deposition have been considered: (i) two mono-nuclear and (ii) one bi-nuclear copper-oxo and copper-hydroxo clusters. To reduce the complexity of the system, due to many possible combinations to couple the unpaired electrons of the Cu(II)/Cu(II) atoms, we decided to study only the deposition of the high-spin ferromagnetically coupled Cu(II)-Cu(II) pairs with a spin multiplicity of 3. Because of the different stoichiometries of the resulting structures, the comparison is done by computing the formation energies of the $Cu_2O_x(OH)_y(H_2O)_z$ with the equation: $\Delta E^{form} = E(Cu\text{-}MOF\text{-}808) - E(MOF\text{-}808) + mE(H_2O) - E(MOF\text{-}808) - nE(precursor)$, where E are the energies of the Cu-MOF-808, MOF-808, $H_2O$, and the copper precursor molecules, and m, n are the number of water and precursor molecules in the formation reaction respectively. As Cu(II) precursor, a molecule with the stoichiometry $Cu(OH)_2(H_2O)_2$ is considered.

As a final step, after obtaining the most stable configurations for the deposited Cu(II) atoms, the interactions with $NO_2$ have been computed. Several initial configurations have been considered, their geometries have been optimised, and the interaction energies with $NO_2$ are computed with the equation: $I.E = E(NO_2\text{-}Cu\text{-}MOF\text{-}808) - E(Cu\text{-}MOF\text{-}808) - E(NO_2)$, where E are the energies of the $NO_2$ complex with Cu-MOF-808, Cu-MOF-808 and $NO_2$, respectively. The total spin multiplicity of the complexes is considered to be a quartet.

During all geometry optimisations, some restrictions have to be applied in order to mimic the crystal environment. Two (2) of the zirconium atoms at the edges of the molecular cluster, and twenty-four (24) oxygen atoms that belong to the ligands are kept frozen. The r2-SCAN-3c functional in combination with the def2-mTZVPP basis set have been used for all geometry optimisations. This low-cost density functional has been shown to perform very well for open-shell transition metal reactions[33,34]. Finally, single point energies with the M06L functional in combination with def2-TZVPP have been performed at the r2-SCAN-3c optimised geometries[35]. All calculations have been performed using the ORCA 5.0.3 programme[36]. Tight criteria have been used for all geometry optimisations and for the convergence of the electronic energies during the SCF.

The analysis of the bonding and non-bonding interactions of $NO_2$ with the MOF-808 and Cu-MOF-808 models is performed by using the Interaction Region Indicator (IRI) method[30]. The population analysis is done by Bader's Quantum Theory of Atoms In Molecules (QTAIM) scheme[37]. The QTAIM and IRI methods are implemented within the Multiwfn 3.8(dev) wavefunction analysis code[38]. The plots have been made with the VMD 1.9.4 software[39].

## Sensing measurements

Room temperature sensing measurements in a flow of nitrogen ($N_2$) and 50 ppm $NO_2$ in synthetic air were performed in a homemade chamber. The chamber has a removable sample loader that the powder places in aluminium support behind a gas-permeable copper TEM grid and two quartz windows on both sides, allowing light to hit the sample. The gas flow could be controlled via flow metres equipped with valves. The materials were photoexcited by a TEEM Photonics Nd:YAG laser ($\lambda = 355$ nm), delivering pulses of 300 ps duration at repetition rates from single shot to 1 kHz. Photoluminescence was collected under 90° by means of an Acton Research SP2500 spectrometer ($f = 500$ mm) equipped with a Princeton Instruments Spec-10 liquid nitrogen cooled back-illuminated deeply depleted CCD for the acquisition of PL spectra. The scattered light arising from the excitation line was cut by placing a 370 nm long-pass filter in front of the spectrometer.

## Data availability

The data that support the findings of this study are available within the article and its Supplementary Information. The source data are available from the corresponding authors upon request. The cartesian coordinates of the optimised computational models and their r2-SCAN-3c and M06-L absolute energies are provided as a separate Supplementary Data 1 file.

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

## Acknowledgements

We thank the Spanish Ministry of Science and Innovation MCIN/AEI/10.13039/501100011033 (grants No. PID2021-123839OB-I00, RYC2018-024328-I and CEX2018-000805-M for A.E.P.-P., grant No. RTI2018-101049-B-I00 for A.M., grants No. RTI2018-097508–B-I00, CEX2020-001039-S and PID2021-128313OB-I00 for J.C.-G., grant No. PID2019-106315RB-I00 for V.A.d.I.P.O.), the "European Union NextGenerationEU/PRTR" (grant No. EUR2020-112294 for A.E.P.-P., grants No. PLEC2021-007906 and TED2021-130173B-C43 for V.A.d.I.P.O.), and the Regional Government of Madrid (TALENTO grants No. 2017-T1/AMB-5264 and 2021-5 A/AMB-20946 for A.M., grant No. S2018/NMT-4511 NMAT2D-CM for J.C.-G., Proyectos Sinérgicos de I + D grant No. Y2018/NMT-5028 FULMATEN-CM for J.C.-G., grant No. S2018/NMT-4367 FotoArt-CM for V.A.d.I.P.O.). V.A.d.I.P.O. acknowledges financial support from the European Research Council (ERC), through the HYMAP project (grant agreement No. 648319), under the European Union's Horizon 2020 research and innovation programme. I.d.C.–V. and I.R.–M. acknowledge FPI-UAM fellowships from Universidad Autónoma de Madrid. C.C.-B. acknowledges the European Social Funds and the Regional Government of Madrid for a postdoctoral contract (PEJD-2018-POST/IND-7909). A.S. acknowledges financial support from the Margarita Salas programme (grant No. CA1/RSUE/2021-00622). Part of this research was carried out at PO2.1 beamline at DESY, a member of the Helmholtz Association (HGF)., and utilising the computing facilities of CSUC. We acknowledge DESY (Hamburg, Germany), a member of the Helmholtz Association HGF, for the provision of experimental facilities. Parts of this research were carried out at PETRA III, and we would like to thank Dr. Welter for his assistance in using the P65 beamline. Beamtime was allocated for proposal I-20210304 EC.

## Author contributions

I.d.C.-V. and C.C.-B. synthesised and characterised the samples. A.S. and J.C.-G. designed and performed the optical sensing experiments. I.R.-M. performed and interpreted the XAS experiments. A.S.J.M.

collected the total scattering data. I.d.C.-V. and A.E.P.-P. analysed and interpreted the PDF data. A.M. carried out the theoretical calculations. F.E.O. and V.A.P.P. performed and interpreted the XPS experiments. I.d.C.-V., A.S., A.M. and A.E.P.-P. wrote the manuscript and all authors contributed to the final version. A.M. and A.E.P.-P. supervised the project.

## Competing interests

The authors declare no competing interests.
