## [Peer Review File · Nature Communications]

Synergistic Binding Sites in a Metal-Organic Framework for the Optical Sensing of Nitrogen DioxideReviewers' Comments:

Reviewer #1:

Remarks to the Author:

The manuscript entitled "Synergistic Binding Sites in a Metal-Organic Framework for the Optical Sensing of Nitrogen Dioxide" is dealing with very important industrial problem for sensing harmful gases like NO₂. Their research combines different experimental and theoretical techniques, which make the report more complete and the conclusions more convincing. The synergism between the theory and experiment is well described, and the data is discussed in a complementary manner. MOFs are indeed very promising materials in terms of their tunability toward various applications. Important elements to consider in optimizing the performance and utility of chemical sensors are sensitivity, selectivity, response time, materials stability, and reusability. While sensitivity depends in part on the method of signal transduction, it also depends on the strength of analyte binding to the MOF (stronger binding translates into lower detection limits) and on the dynamics of analyte transport within the MOF. In the current study the authors pointed to some of the important characteristic of the sensors: sensitivity, materials stability, and reusability. They still need to answer one important questions. What is the selectivity of the material with respect to other gas molecules, which may coexist as impurities in the NO₂ supposed to be detected? In NO₂ gas may exist also water vapors or other NO_x and if they bind stronger and irreversibly to the active sites, they will "poison" the sensing ability. This is very important question, which should be addressed. Indeed, the material proposed by the authors show sensing activities at significant low NO₂ concentrations, down to 10 ppm. How does it compare with other NO₂ sensors? What is the benefit of their material over the others NO₂ sensors? There are MOFs, for instance M-MOF-74, which are capable to detect NO₂ at even lower concentration (5 ppm).

The local interaction of the NO₂ with the framework (with or without Cu) is discussed in details and based mainly on DFT modeling. In order to clarify the NO₂ interaction with the MOF, the authors may discussed another important points. For example the DFT results suggest charge transfer of 0.23 e, from Cu-sites to the HOMO of NO₂ molecule. The HOMO of NO₂ molecule exhibit anti-bonding character, so such charge transfer should affect the N-O bond strength. In this respect the FT-IR of NO₂ loaded MOF and free NO₂ should show shift in the N-O vibration upon sorption on the Cu-sites. The authors provided FT-IR for the MOF samples only, but not the one loaded with NO₂ and such analysis is not possible from the presented results. Such comparison will complement the DFT results and will bring deeper understanding for the nature of the adsorbent-adsorbate interactions and state of the Cu sites. Usually FT-IR technique of probe molecules (CO, NO, MeOH etc) is well established technique to analyze the strength of Lewis acid and base sites in porous materials (zeolites, MOFs) or metal-oxide surfaces.

There are some technical issues, which should be also addressed in the main text and in the SI as well. In the main text on Fig 5 and in the SI on Fig S11.1 S11.2, the H atoms are represented in white color on white background. By this reason it is difficult for the reader to see the positions of the H atoms. Thus the proposed cluster models are unreadable and unclear. In the SI section "Results", page 18, the authors say "Subsequently, based on the observation for the deposition of one copper atom, several possible conformations have been considered for the deposition of two copper atoms." The term "conformation" in chemistry is related to the isomerism due to rotation of functional groups or part of the molecule around single bond. Is the term "conformation" used correctly here? I would rephrase the sentence using "configuration" instead of "conformation", because on fig S11.1 are shown structures with different mutual arrangements of the two Cu ions.

In conclusion, the manuscript presents successfully incorporated synergistic binding sites in MOF-808, resulting in excellent optical sensing properties towards NO₂ at remarkably low concentrations. The combination of experimental techniques with computational modeling is used to get insight in to the atomistic structure and host-guest interactions in the system under study. This work opens new prospects in the field of optical sensing of toxic gases using MOF materials by tuning the host-guest interactions through post-synthetic chemical modifications. If the authors improve the manuscript and the SI answering the questions raised above, I would suggest the manuscript as suitable for publication in Nature communication journal.

Reviewer #2:

Remarks to the Author:

This manuscript discusses the synthesis and characterization of a Cu-MOF-808 structure, based on post-synthetic processing of the MOF. The MOF-808 structure remains even after incorporation of the Cu, and appears to be stable, with an even distribution of Cu. The composition of the metal cluster varies with the type of Cu-precursor used, with a maximum Cu concentration of 3.3 per Zr-cluster. Limited changes in the structure were noted, along with increased adsorption of NO₂ accompanied by changes in luminescence.

The authors have not made a compelling case that the MOF structures are demonstrating a novel or unique capability relative to other MOF structures with known NO₂ sensing capabilities, even at low (ppm/ppb) concentration or low temperature, see some examples below. That said, the incorporation of Cu into MOF-808 is unique but its particular application in this area is less clear.

- <https://pubs.acs.org/doi/full/10.1021/acsami.0c10420>
- <https://pubs.acs.org/doi/full/10.1021/acsami.0c00803>
- <https://onlinelibrary.wiley.com/doi/full/10.1002/adfm.202006598>
- <https://www.sciencedirect.com/science/article/pii/S0925400520314003>

Simulations are used to predict the stable locations of the Cu in the metal clusters of the Cu-MOF-808 structure based on a simplified gas-phase model of the system composed of two metal clusters linked by two ligands. The authors have used previously used similar clusters for evaluation of these MOF systems. The methodology used to evaluate the formation enthalpy is typically used in literature, where the energy balance of the reaction is used to predict the thermodynamic stability of the product. Overall, the conclusions are supported by the presented evidence and the methodology is sound with sufficient detail to replicate the results. However, there are several comments that can be addressed:

1. The replacement of the Zr by the Cu justifies the evaluation of the minimized cluster model. Either 1 or 2 Cu atoms in the cluster model are evaluated for their binding. In the experimental portion, it was proposed that there is quite a bit more Cu metal clusters in the model. Why were high numbers of Cu clusters included in the model? That would be more representative of the structure itself and any distortion that may be arising from this high concentration of Cu per cluster.
2. The structural analysis seems to indicate that in some cases there may be sufficient deposited Cu in the pore structure to entirely fill some of the smaller pore spaces (seen by the decrease in surface area for the Cu-MOF-808 data). Since the NO₂ looks to be binding primarily with the Cu what does the MOF-808 structure provide for NO₂ binding/sensing? Is it serving primarily as a scaffold? Would this process exist in other MOF structures?
3. On a more methodological note, it is not clear why Cu(OH)₂(H₂O)₂ is selected as a precursor. If the intention was to use hydrated Cu²⁺ ions then a five-fold coordination is more favored (<https://www.science.org/doi/10.1126/science.291.5505.856>).
4. There was a missed opportunity to use excited state calculations to directly probe the impact of the NO₂ adsorption on the luminescence properties of the Cu-Zr metal clusters. In particular, the charge transfer to the LUMO orbital is the main comment on the source of the change in the optical properties. This appears to be more of a side comment, and a more thorough investigation of the source of the optical changes would greatly strengthen the impact of the manuscript.

Reviewer #3:

Remarks to the Author:

This manuscript by Mavrandonakis, Platero-Prats and co-workers reports the incorporation of Cu in the Zr-based MOF-808 by post-functionalization, its characterization by various techniques including Pair

Distribution Function and X-ray absorption spectroscopy and its luminescent properties that can be exploited to obtain an optical sensor for the detection of low concentrations of NO₂. I do not recommend the publication of this paper in Nature Communications for the following reasons:

1) The first reason is the lack of originality : this work follows two previous reports from the same group that are closely related, the first one on Fe-oxo clusters stabilized on MOF-808 (ref. 15) and the second one on MOF-808 decorated with single copper sites (ACS Applied Mater. Interfaces 2022, 14, 27040, not cited). The originality of the paper could lie in the performances of the material as sensor.

This property has not been previously investigated in the two previous reports. However the significance of this property must be shown. Could the authors compare the performance of their material to other materials reported in the literature? Furthermore, what is the selectivity of this material for NO₂ compared to other gases ?

3) My second main concern is about the chemical formula proposed in the experimental section: [Zr₆Cu_{3.308}(C₉H₃O₆)₂(H₂O)₂₅]. This formula obviously does not respect the electroneutrality. There is clearly a lack of negative charges. C, H, N analysis must be given.

4) Finally the paper is confusing about the position of the Cu ions in the MOF and the agreement of DFT, EXAFS and DFT calculations about the Cu···M interactions. For example it is written page 6 line 120 that "According to both the EXAFS data and the differences of X-ray scattering power of Cu and Zr, the most plausible assignment is Cu···Zr". However the existence of Cu···Zr interactions is not mentioned in the EXAFS section. It is also written page 7 line 45 that "The characteristics, i.e the distorted planar coordination environment of the copper atom and the Cu-Cu and Cu···Zr distances, for models B and C indicate very good agreement with the experimental PDF/EXAFs data." However PDF and EXAFS data do not show the existence of Cu-Cu interactions.

Minor corrections:

1) There should be a more detailed bibliographic paragraph on studies of the introduction of Cu or other metal sites in Zr-based MOFs reported in the literature. For example, the paper published in 2019 in Nature Catalysis of Bing An et al. (doi 10.1038/s41929-019-0308-5) must be cited.

2) The authors indicate page 5 line 108 that the peak centered at ca. 1.6 Å (value without phase correction) corresponds to Cu(II)-O bonds. Could they give the value after correction for non-specialists ? Does it correspond to the value found by PDF ?

3) Figure 5 is not clear enough, it is not easy to see the coordination sphere around the copper ions. Are there OH groups connected to Zr ions ? In this case why do not they appear in the formula (the lack of negative charge would thus be even larger) ?

4) What is the concentration of gas in Figure 6A ? This must be indicated in the legend of Figure 6.

5) In Figure 6A, there seems to be a decrease in the response of Cu-MOF-808 with time so that the term "large reversibility through N₂ purging" seems a bit exaggerated.

Reviewer 1

Comments to the Author

Summary

The manuscript entitled "Synergistic Binding Sites in a Metal-Organic Framework for the Optical Sensing of Nitrogen Dioxide " is dealing with very important industrial problem for sensing harmful gases like NO₂. Their research combines different experimental and theoretical techniques, which make the report more complete and the conclusions more convincing. The synergism between the theory and experiment is well described, and the data is discussed in a complementary manner.

MOFs are indeed very promising materials in terms of their tunability toward various applications.

Comments

1. Important elements to consider in optimizing the performance and utility of chemical sensors are sensitivity, selectivity, response time, materials stability, and reusability. While sensitivity depends in part on the method of signal transduction, it also depends on the strength of analyte binding to the MOF (stronger binding translates into lower detection limits) and on the dynamics of analyte transport within the MOF. In the current study the authors pointed to some of the important characteristic of the sensors: sensitivity, materials stability, and reusability. They still need to answer one important questions. What is the selectivity of the material with respect to other gas molecules, which may coexist as impurities in the NO₂ supposed to be detected? In NO₂ gas may exist also water vapors or other NO_x and if they bind stronger and irreversibly to the active cites, they will "poison" the sensing ability. This is very important question, which should be addressed.

This is indeed a very interesting point. We have carried out new experiments to determine the limit of detection of both MOF-808 and Cu-MOF-808 to NO₂ and other relevant gases found in air such as CO₂ and O₂. These gases may be able to potentially bind to the copper sites and poison the optical response of this material. Interestingly, our results indicate significantly higher sensitivity of **Cu-MOF-808 towards NO₂ (0.016 ppm)** than CO₂ (2837 ppm) and O₂ (476 ppm). In the case of MOF-808, we have observed a different behaviour to sense CO₂, followed by a subtle PL enhancement which makes difficult to determine the limit of detection (LOD).

The LOD values were calculated based on the standard deviation (σ) obtained from measuring the PL spectra of the materials under N₂ (0 ppm NO₂) at fourteen separate spectra. Then, after linear fitting of the materials under exposure to different gases, the slopes were incorporated in the following formula:

$$\text{LOD} = 3.3 \sigma / \text{slope}$$

Table 1. LOD values of MOF-808 and Cu-MOF-808 towards NO₂, CO₂ and O₂.

Material	LOD (NO ₂)	LOD (CO ₂)	LOD (O ₂)
MOF-808	0.101 ppm	-	588 ppm
Cu-MOF-808	0.016 ppm	2837 ppm	476 ppm

We have now added the following sentence in the main text:

Page 9 – “...longer respond of NO₂ gas. The materials were then exposed to different concentrations of NO₂ to assess the limit of detection (LOD), resulting in values of 101 and 16 ppb for MOF-808 and Cu-MOF-808, respectively. Furthermore, the sensitivity of the Cu-MOF-808 to other common gases present in air potentially able to bind to the copper sites were explored, such as CO₂ and O₂. The results confirmed a significantly higher sensitivity of Cu-MOF-808 to NO₂ compared to CO₂ and O₂ (see Figure S12.6). These results demonstrate the best optical response reported so far for a MOF material towards the sensing of NO₂”.

We have also added two new figures in the SI (Figure S13.6-7) to show the sensitivity of MOF-808 and Cu-MOF-808 to NO₂, CO₂ and O₂.

Fig. S13.6. Linear fitting the PL quenching of MOF-808 and Cu-MOF-808 exposed to different concentrations of NO₂ (0 – 10 ppm).

Fig. S13.7. PL quenching of MOF-808 and Cu-MOF-808 exposed to different concentrations of NO₂, CO₂ and O₂ after activating with N₂ (λ_{exc}: 355 nm).

We have also explored computationally the nature of the binding between competing molecules (such as H₂O, NO, CO and CO₂) with the Cu-MOF-808. The relative strength of the interactions between the competing molecules (NO₂, NO, CO₂, CO and H₂O) and the copper site is used as a qualitative descriptor to predict, which of the above molecules will be preferentially adsorbed. However, recent experimental and computational studies (*Chem. Mater.* **2022**, 34, 17, 7906–7915 and *Inorg. Chem.* **2023**, 62, 2, 950–956) have shown that the binding strength does not always indicate the preferred adsorbed molecule, and that a combination/competition between thermodynamics (binding energies) and kinetics (energy barriers) defines the adsorbed molecule. For example, although NH₃ has stronger binding than H₂O with the open metal sites of the Ni- and Mg-MOF-74, the presence of multiple H₂O molecules will cause the displacement of a preabsorbed NH₃ molecule (*Chem. Mater.* **2022**, 34, 17, 7906–7915). Moreover, the displacement of CO bound inside Ni-MOF-74 (binding energy of 53 kJ/mol) is readily driven by CO₂ exposure, even though CO₂ has a noticeably weaker binding energy of 41 kJ/mol (*Inorg. Chem.* **2023**, 62, 2, 950–956).

Here, we have assessed the competitive adsorption of NO₂, NO, CO₂, CO and H₂O by comparing their interaction energies with the copper sites. The copper sites have distorted square-planar geometries. Therefore, the binding site of the adsorbed molecule is considered to be on the axial position.

Table 2. Interaction energies (in kJ mol⁻¹) and distances (in Å) of NO₂, NO, CO₂ and H₂O interacting the copper sites of model-C. In model C, two copper sites with slightly different coordination environment are present.

Model-C-2Cu	I.E. (kJ mol ⁻¹) ^a	Charge Transfer MOF-800 molecule (AIM) ^b	Cu- →	I.E. (kJ mol ⁻¹) ^a	Charge Transfer MOF-800 molecule (AIM) ^b	Cu- →
NO ₂	-59.1	+0.23		-48.5	+0.16	
NO	-43.6	+0.01		-38.3	+0.03	
CO ₂	-50.0	+0.01		-41.2	+0.01	
CO	-44.8	-0.01		-39.5	+0.01	
H ₂ O	-66.9 (first ^c) -66.3 (second ^c) -73.1 (third ^c)	n.c		-77.3	0.00	

^a Two values for the interaction energies are reported, because two copper sites with slightly different coordination environment are present.

^b Positive value for the computed charge transfer means that charge is flowing from the MOF towards the adsorbed molecule.

^c The interaction of up to three H₂O molecules with the copper site has been investigated in this case. In all of these three cases, the water molecule moves away from the copper site and interacts with hydrogen-bonds with the neighbouring hydroxo and aqua ligands of the ZrO₂ node.

In the cases of NO and CO, the interaction energies with the Cu(II) sites are weaker than the interaction of NO₂. This is consistent with previous works by us (*J. Phys. Chem. C* **2013**, 117, 14570-14578) and others (*Nanomaterials* **2018**, 8, 958) on the CO and NO_x binding with the open metal sites of the HKUST-1 MOF. This is not unexpected, considering that NO and CO interact stronger with Cu(I) sites than the Cu(II) due to significant pi-backdonation. (*J. Phys. Chem. C* **2018**, 122, 30, 17238–17249) The interaction energies of CO₂ with the two Cu(II) sites are computed weaker compared to NO₂, with values of -50.0 and -41.2 kJ mol⁻¹. Unexpectedly,

the preferred adsorption site of the water molecule is not on the copper site. During the geometry optimization, the water molecule moves from the axial site of the Cu(II) towards the ZrO₂ node and prefers to interact via Hydrogen-bonds with the hydroxo and aqua ligands. Same results are obtained, when a second and third water molecule are inserted on top of the copper site. In all cases, the water molecules move away from the axial binding site of the Cu(II). Although, the interaction energies of H₂O with the Cu(II) sites are calculated to be stronger than of NO₂, the presence of water molecule will not affect the sensing, because the axial site of one copper centre is still available for binding with NO₂ in the presence of water molecules. In the second copper site, the water molecule remains adsorbed on the metal axial position with an interaction energy of -77.3 kJ mol⁻¹ that is significantly stronger than the computed interaction energy of NO₂ (-48.5 kJ mol⁻¹). Thus, at humid conditions one of the two copper sites will still be available for adsorbing and sensing NO₂.

In conclusion, the calculations suggest that **NO, CO and CO₂ are adsorbed less strongly than the NO₂, and that at humid conditions one of the two copper sites will be able to bind NO₂.**

We have added the above-mentioned results in page 13 of the manuscript and the end of the section S12 in the SI.

2. Indeed, the material proposed by the authors show sensing activities at significant low NO₂ concentrations, down to 10 ppm. How does it compare with other NO₂ sensors? What is the benefit of their material over the others NO₂ sensors? There are MOFs, for instance M-MOF-74, which are capable to detect NO₂ at even lower concentration (5 ppm).

We thank the referee for giving us the opportunity to expand on this point. Our new results show that the incorporation of synergistic copper sites into the Zr₆O₈ cluster of MOF-808 improves the capture of NO₂ at the sub-ppm level. According to our last results, the limit of detection (LOD) for Cu-MOF-808 and MOF-808 was estimated to be 16 ppb and 101 ppb, respectively. While there are reported works that show the performance of some MOFs to detect NO₂ gas at very low concentrations (at ppb level), their sensing mechanism are not similar to the one reported in this work. As shown in the following table, the most common methods to detect NO₂ are based on chemoresistivity, and organic field-effect transistors (OFETs) that measure electrical resistance and electrical conductivity. Optical sensing is a fast and simple way to detect NO₂ gas compared to other methods. The development of novel materials with unique optical properties towards the detection of NO₂ at sub-ppm level can afford the fabrication of cost-effective and portable devices.

We have now included the following table in the SI (Section S12), showing a comparative analysis of NO₂ sensors reported in the literature.

Table S13.1. Comparison of different materials employed as NO₂ sensors reported in the literature.

Material	Sensor type	T (°C)	Concentration, LOD	Response	Ref.
In/ZnO-10	Chemoresistive	300	0.20 ppb	Electrical resistance	[1]
[Ni(TPyP)-(TiF ₆)]@PDVT-10	OFET/ Chemoresistive	25	8.25 ppb	Electrical resistance	[2]
Co ₃ O ₄ /Biomass carbon	Chemoresistive	25	10 ppb	Electrical conductivity	[3]
In ₂ O ₃ /ZIF-8	Chemoresistive	140	10 ppb	Electrical resistance	[4]
Cu ₃ (HHTP) ₂ /Fe ₂ O ₃	Chemoresistive	20	11 ppb	Electrical resistance	[5]
CuO tube-like nanofibers	Chemoresistive	25	12.2 ppb	Electrical resistance	[6]
PDVT-10	OFET/ Chemoresistive	25	25 ppb	Electrical resistance	[7]
Ti-FIR-120	Chemoresistive	25	40 ppb	Electrical resistance	[8]
Pt@Cu ₃ (HHTP) ₂ thin-film	Chemoresistive	25	0.1 ppm	Electrical resistance	[9]
Cu ₃ (HHTP) ₂ thin-film	Chemoresistive	25	0.1 ppm	Electrical resistance	[9]
Cu-MOF250 (unknown structure)	Chemoresistive	40	0.14 ppm	Electrical resistance	[10]
Ag-3 doped-WO ₃	Chemoresistive	30	196.8 ppb	Electrical resistance	[11]
Ni-MOF-74	Capacitive	50	0.5 ppm	Electrical resistance	[12]
Pd@Cu ₃ (HHTP) ₂ thin-film	Chemoresistive	25	1 ppm	Electrical resistance	[13]
Pt@Cu ₃ (HHTP) ₂ thin-film	Chemoresistive	25	1 ppm	Electrical resistance	[13]
NiFe ₂ O ₄ nanofibers	Chemoresistive	350	10 ppm	Electrical resistance	[14]
((H ₃ O) ₄ [Co ₂ (L)(DMF)(H ₂ O) ₄] ₂ DMF·3H ₂ O)	Chemoresistive	25	-	Electrical conductivity	[15]
ZIF-8	Chemoresistive	350	-	Electrical resistance	[16]
Porphyrin LB Films	Optical	25	0.46 ppm	Red-shift of UV-Vis spectrum	[17]
TCPC@MOF@PDMS	Optical	25	> 0.5 ppm	PL quenching	[18]
Porphyrin-film@SiO ₂	Optical	25	1.0 ppm	UV-Vis peaks shift/Intensity change and turn-on absorption	[19]
{[Tb ₂ (NBDC) ₃ (DMF) ₄]2DMF}	Optical	25	1.8 ppm	PL quenching	[20]
{[Eu ₂ (NBDC) ₃ (DMF) ₄]2DMF}	Optical	25	2.2 ppm	PL enhancement	[20]
Tb(BTC)@PDMS	Optical	25	4 ppm	PL quenching	[21]
Calixarene-based Zr-MOF film	Optical/ Chemoresistive	25	5 ppm	Colorimetric/ Turn-on UV-vis absorption/photo-current change	[22]
Zn-ZJU-66 film	Optical	25	-	PL quenching	[23]
Y-DOBDC	Optical	25	-	Colorimetric/ PL quenching	[24]
((H ₃ O) ₄ [Co ₂ (L)(DMF)(H ₂ O) ₄] ₂ DMF·3H ₂ O)	Optical/ Chemoresistive	25	-	Colorimetric/ Conductivity	[25]

MOF-808	Optical	25	101 ppb	PL quenching	In this work
Cu- MOF-808	Optical	25	16 ppb	PL quenching	In this work

References:

- Li, Z. Zhang, Y. Zhang, H. Jiang, Y. and Yi, J. Superior NO₂ sensing of MOF-derived indium-doped ZnO porous hollow cages. *ACS Appl. Mater. Interfaces*, **12**, 37489-37498 (2020).
- Yuvaraja, S. Surya, S.G. Chernikova, V. Vijjapu, M.T. Shekhah, O. Bhatt, P.M. Chandra, S. Eddaoudi, M. and Salama, K.N. Realization of an ultrasensitive and highly selective OFET NO₂ sensor: the synergistic combination of PDVT-10 polymer and porphyrin–MOF. *ACS Appl. Mater. Interfaces*, **12**, 18748-18760 (2020).
- Chen, J, Lv, H. Bai, X. Liu, Z. He, L. Wang, J. Zhang, Y. Sun, B. Kan, K. and Shi, K. Synthesis of hierarchically porous Co₃O₄/Biomass carbon composites derived from MOFs and their highly NO₂ gas sensing performance. *Microporous Mesoporous Mater*, **321**, 111108 (2021).
- Liu, Y. Wang, R. Zhang, T. Liu, S. and Fei, T. Zeolitic imidazolate framework-8 (ZIF-8)-coated In₂O₃ nanofibers as an efficient sensing material for ppb-level NO₂ detection. *J. Colloid Interface Sci*, **541**, 249-257 (2019).
- Jo, Y.M. Lim, K. Yoon, J.W. Jo, Y.K. Moon, Y.K. Jang, H.W. and Lee, J.H. Visible-light-activated Type II heterojunction in Cu₃(hexahydroxytriphenylene)₂/Fe₂O₃ hybrids for reversible NO₂ sensing: Critical role of π – π^* transition. *ACS Cent. Sci.*, **7**, 1176-1182 (2021).
- Liu, J. Wang, W. Li, G. Bian, X. Liu, Y. Zhang, J. Gao, J. Wang, C. Zhu, B. and Lu, H. Metal–organic framework-derived CuO tube-like nanofibers with high surface area and abundant porosities for enhanced room-temperature NO₂ sensing properties. *J. Alloy. Comp.* **934**, 167950 (2023).
- Yuvaraja, S. Surya, S.G. Vijjapu, M.T. Chernikova, V. Shekhah, O. Eddaoudi, M. and Salama, K.N. Fully Integrated Organic Field-Effect Transistor Platform to Detect and to Quantify NO₂ Gas. *Phys. Status Solidi RRL*, **14**, 2000086 (2020).
- Li, H.Z. Pan, Y. Li, Q. Lin, Q. Lin, D. Wang, F. Xu, G. and Zhang, J. Rationally designed titanium-based metal–organic frameworks for visible-light activated chemiresistive sensing. *J. Mater. Chem. A*, **11**, 965 (2023).
- Kim, J.O. Koo, W.T. Kim, H. Park, C. Lee, T. Hutomo, C.A. Choi, S.Q. Kim, D.S. Kim, I.D. and Park, S. Large-area synthesis of nanoscopic catalyst-decorated conductive MOF film using microfluidic-based solution shearing. *Nat. Commun.* **12**, 4294 (2021).
- Arul, C. Moulae, K. Donato, N. Iannazzo, D. Lavanya, N. Neri, G. and Sekar, C. Temperature modulated Cu-MOF based gas sensor with dual selectivity to acetone and NO₂ at low operating temperatures. *Sensor. Actuat. B-Chem.* **329**, 129053 (2021).
- Mathankumar, G. Harish, S. Mohan, M.K. Bharathi, P. Kannan, S.K. Archana, J. and Navaneethan, M. Enhanced selectivity and ultra-fast detection of NO₂ gas sensor via Ag modified WO₃ nanostructures for gas sensing applications. *Sensor. Actuat. B-Chem.* 133374 (2023).

12. Small, L.J. Henkelis, S.E. Rademacher, D.X. Schindelholz, M.E. Krumhansl, J.L. Vogel, D.J. and Nenoff, T.M. Near-zero power MOF-based sensors for NO₂ detection. *Adv. Funct. Mater.* **30**, 2006598 (2020).
13. Koo, W.T. Kim, S.J. Jang, J.S. Kim, D.H. and Kim, I.D. Catalytic metal nanoparticles embedded in conductive metal–organic frameworks for chemiresistors: highly active and conductive porous materials. *Adv. Sci.* **6**, 1900250 (2019).
14. Van Hoang, N. Hiep, N.T. Hung, N.M. Nguyen, C.V. Hung, P.T. Hoat, P.D. and Heo, Y.W. Optimization of synthesis conditions and sensing performance of electrospun NiFe₂O₄ nanofibers for H₂S and NO₂ detection. *J. Alloy. Comp.* **936**, 168276, (2023).
15. Ma, Y.X. Gao, B. Li, Y. Wei, W. Zhao, Y. and Ma, J.F. Macrocyclic-Based Metal–Organic Frameworks with NO₂-Driven On/Off Switch of Conductivity. *ACS Appl. Mater. Interfaces*, **13**, 27066–27073 (2021).
16. Zhan, M. Hussain, S. AlGarni, T.S. Shah, S., Liu, J. Zhang, X. Ahmad, A. Javed, M.S. Qiao, G. and Liu, G. Facet controlled polyhedral ZIF-8 MOF nanostructures for excellent NO₂ gas-sensing applications. *Mater. Res. Bull.* **136**, 111133 (2021).
17. Pedrosa, J.M. Dooling, C.M. Richardson, T.H. Hyde, R.K. Hunter, C.A. Martin, M.T. and Camacho, L. Characterization and fast optical response to NO₂ of porphyrin LB films. *Mater. Sci. Eng. C*, **22**, 433–438 (2002).
18. Queirós, C. Moscoso, F.G. Almeida, J. Silva, A.M. Sousaraei, A. Cabanillas-González, J. Ribeiro Carrott, M. Lopes-Costa, T. Pedrosa, J.M. and Cunha-Silva, L. MOF-Based Materials with Sensing Potential: Pyrrolidine-Fused Chlorin at UiO-66 (Hf) for Enhanced NO₂ Detection. *Chemosensors* **10**, 511 (2022).
19. Gulino, A. Mineo, P. Scamporrino, E. Vitalini, D. and Fragalà, I. Molecularly engineered silica surfaces with an assembled porphyrin monolayer as optical NO₂ molecular recognizers. *Chem. Mater.* **16**, 1838–1840 (2004).
20. Gamonal, A. Sun, C., Mariano, A.L. Fernandez-Bartolome, E. Guerrero-SanVicente, E., Vlaisavljevich, B. Castells-Gil, J. Marti-Gastaldo, C. Poloni, R. Wannemacher, R. and Cabanillas-Gonzalez, J. Divergent adsorption-dependent luminescence of amino-functionalized lanthanide metal–organic frameworks for highly sensitive NO₂ sensors. *J. Phys. Chem. Lett.* **11**, 3362–3368 (2020).
21. Moscoso, F.G. Almeida, J. Sousaraei, A. Lopes-Costa, T. Silva, A.M. Cabanillas-Gonzalez, J., Cunha-Silva, L. and Pedrosa, J.M. Luminescent MOF crystals embedded in PMMA/PDMS transparent films as effective NO₂ gas sensors. *Mol. Syst. Des. Eng.* **5**, 1048-1056 (2020).
22. Schulz, M. Gehl, A. Schlenkrich, J. Schulze, H.A. Zimmermann, S. and Schaate, A. A Calixarene-Based Metal–Organic Framework for Highly Selective NO₂ Detection. *angew. Chem. Int. Ed.* **57**, 12961–12965 (2018).
23. Zhang, J. Hu, E., Liu, F. Li, H. and Xia, T. Growth of robust metal–organic framework films by spontaneous oxidation of a metal substrate for NO₂ sensing. *Mater. Chem. Front.* **5**, 6476 (2021).
24. Sava Gallis, D.F. Vogel, D.J. Vincent, G.A. Rimsza, J.M. and Nenoff, T.M. NO_x adsorption and optical detection in rare earth metal–organic frameworks. *ACS Appl. Mater. Interfaces*, **11**, 43270–43277 (2019).

25. Ma, Y.X. Gao, B. Li, Y. Wei, W. Zhao, Y. and Ma, J.F. Macrocycle-Based Metal–Organic Frameworks with NO₂-Driven On/Off Switch of Conductivity. *ACS Appl. Mater. Interfaces*, **13**, 27066–27073 (2021).

3. The local interaction of the NO₂ with the framework (with or without Cu) is discussed in details and based mainly on DFT modeling. In order to clarify the NO₂ interaction with the MOF, the authors may be discussed another important points. For example the DFT results suggest charge transfer of 0.23 e, from Cu-sites to the HOMO of NO₂ molecule. The HOMO of NO₂ molecule exhibit anti-bonding character, so such charge transfer should affect the N-O bond strength. In this respect the FT-IR of NO₂ loaded MOF and free NO₂ should show shift in the N-O vibration upon sorption on the Cu-sites. The authors provided FT-IR for the MOF samples only, but not the one loaded with NO₂ and such analysis is not possible from the presented results. Such comparison will complement the DFT results and will bring deeper understanding for the nature of the adsorbent-adsorbate interactions and state of the Cu sites. Usually FT-IR technique of probe molecules (CO, NO, MeOH etc) is well established technique to analyze the strength of Lewis acid and base sites in porous materials (zeolites, MOFs) or metal-oxide surfaces.

This is indeed a very interesting point. We have not been able to probe the Cu⋯NO₂ interactions with FT-IR experiments, because the NO₂ bands are overlapping with vibrations from the organic ligands of the MOF framework (*Nature Mater.* **2018**, *17*, 691–696). However, we have explored the adsorbent-adsorbate interactions by performing X-ray photoelectron spectroscopy (XPS) on the NO₂-loaded Cu-MOF-808. XPS data of the NO₂-loaded Cu-MOF-808 show feature peaks in the Cu 2p region at 934.1 eV-value well within the expected binding energy for Cu(II) cations. The Cu 2p region also includes high binding energy satellites characteristic of Cu(II) in a d⁹ electron configuration. This fact evidences the lack of variations in terms of oxidation state of the copper sites under exposure to NO₂. Interestingly, the N 1s region of the NO₂-treated Cu-MOF-808 can be fitted to two components centred at 407.1 eV and 404.4 eV. Based on literature, the major component at 407.1 eV can be reasonably well assigned to oxidized species of NO₂ adsorbed on the metal-oxide surfaces such as the Zr₆O₈ clusters (e.g. ONO⋯(OH)-Zr(IV) species), (*J. Am. Chem. Soc.* **2001**, *123*, 9597; *Phys. Chem. Chem. Phys.*, **2009**, *11*, 8295), as previously seen in Zr-MOFs under dry conditions (*Langmuir*, **2013**, *29*, *1*, 168). The weak low-binding energy component at 404.4 eV could be assigned to reduced NO₂ species interacting with the Cu-oxo sites. The appearance of this low-binding energy signal demonstrates not only the interaction with the copper sites (charge transfer) but also with the Zr₆O₈ clusters, as suggested by the DFT calculations.

Fig. S11.1. XPS spectrum in the **N1s** for NO_2 -loaded Cu-MOF-808, showing the presence of reduced chemisorbed NO_2 species.

We have now edited the following paragraph in the main text to include this experimental evidence from XPS experiments:

Page 10-11. “To shed light into the relevant sensing mechanism, we first measured the PXRD to understand whether the NO_2 inclusion interfered with the Cu-MOF-808 crystallinity. Figure 6D shows the PXRD data of Cu-MOF-808 before and after NO_2 sensing. FT-IR spectroscopy data collected on Cu-MOF-808 before and after the sensing experiments revealed the absence of major changes (Figure S8.2), showing the stability of the framework after exposure to NO_2 . To gain insight into the local nature of the adsorption of NO_2 within Cu-MOF-808, X-ray photoelectron spectroscopy (XPS) experiments were performed to elucidate the binding of NO_2 to the copper centres. As shown in figure S11.2, XPS spectra of Cu-MOF-808 before and after exposure to NO_2 presents feature peaks in the Cu 2p region at 934.1 eV value well within the expected binding energy for Cu(II) cations. The Cu 2p region also includes high binding energy satellites characteristic of Cu(II) in a d^9 electron configuration. This fact evidences the lack of variations in terms of oxidation state of the copper sites under exposure to NO_2 . Interestingly, the N1s region of the NO_2 -loaded Cu-MOF-808 (Figure S11.1) can be fitted with two components centred at binding energies 407.1 eV and 404.4 eV. The high bind energy component matches closely species of NO_2 adsorbed on metal-oxide surfaces,^{28,29} and therefore we tentatively assigned this component to species formed upon NO_2 adsorption on the Zr_6O_8 centres of the MOF sample. On the other hand, the low binding energy component centred at 404.4 eV could be assigned to partially reduced NO_2 species interacting with the copper-oxo sites, similar to those occurring upon NO_2 adsorption on Cu_2O (111) surfaces.³⁰ Based on or DFT calculations, in the interaction of Cu centres with NO_2 , there is a net transfer of electronic charge to the NO_2 molecule, which leads to a partial reduction, consistent with the observation of a low binding energy component in the N 1s region. Therefore, the incorporation of isolated copper hydroxo sites within the MOF 808 structure improved the optical sensing response against NO_2 at low concentrations and enhanced its reversibility.”

We have also included the following Figures in SI document.

Fig. S11.1. XPS spectrum in the N 1s for NO₂-loaded Cu-MOF-808, showing the presence of reduced chemisorbed NO₂ species.

Fig. 11.2. XPS spectrum in the Cu 2p_{3/2} for NO₂-loaded Cu-MOF-808, showing the presence of Cu(II) sites.

Fig. S11.3. XPS spectrum in the C 1s for NO₂-loaded Cu-MOF-808, showing the presence aliphatic and C=O species.

Motivated by the reviewer comment, we have further analysed the charge transfer mechanism towards NO₂. The population analysis (AIM and also CM5) indicates that the copper partial charge remains the same before and after NO₂ binding. The analysis pointed towards a complex mechanism, where charge is transferred from the MOF framework to the NO₂ through the copper sites. The charge is shifted mainly from the carbon atoms of the carboxylate groups. This charge depletion from the framework atoms can explain the luminescence quenching of the Cu-MOF-808 in the presence of NO₂.

We have edited the manuscript and added the following text in Page 12:

“A charge population analysis based on AIM and also CM5 charge schemes indicates that the copper partial charge remains the same before and after NO₂ binding. The analysis reveals a complex mechanism, where charge is transferred from the MOF framework to the NO₂ through the copper sites. The charge is shifted mainly from the carbon atoms of the carboxylate groups. This charge depletion from the framework atoms can explain the luminescence quenching of the Cu-MOF-808 in the presence of NO₂.”

4. There are some technical issues, which should be also addressed in the main text and in the SI as well. In the main text on Fig 5 and in the SI on Fig S11.1 S11.2, the H -atoms are represented in white color on white background. By this reason it is difficult for the reader do see the positions of the H atoms. Thus, the proposed cluster models are unreadable and unclear.

We have now edited Fig 5 and Fig 7 and in the SI on Fig S12.1 S12.2 by changing the colour of the hydrogen atoms (main text) or the background (SI). We hope the revised structural models are clearer for the readers of the article.

Figure 5. (A) Formation energies (ΔE^{form}) and relative formation energies ($\Delta\Delta E^{\text{form}}$) in kJ/mol for the deposition of two copper-hydroxide species on the MOF-808 nodes. The three most stable configurations are presented here. The formation energies are calculated by considering a reaction of the MOF-808 with two copper precursor species and the subsequent release of water molecules. The relative formation energies are referred versus the formation energy of the model A-2Cu. (B) Structural and (C) schematic representation of the hetero-bimetallic tetrahedral structural subunit in Cu-MOF-808. Colour scheme: blue = Zr, grey = C, magenta = μ_3 -O, orange = Cu, red = O, green = H.

Figure 7. (A) Adsorption geometries of NO_2 in Cu-MOF-808 (left) and MOF-808 (right). **Colour scheme:** blue = Zr, grey = C, magenta = $\mu_3\text{-O}$, orange = Cu, red = O, green = H. (B) Interaction Region Indicator (IRI) plots for NO_2 interacting with Cu-MOF-808 (left) and MOF-808 (right). (C) IRI plots for the Cu-MOF-808 (left) and MOF-808 (right). The adsorption region is highlighted with a rectangular box. The coloring scheme is also explained in ESI (Figure S12.3).

We have also changed Figures S12.1 and S12.2 from the SI. We have changed the background colour to black, thus the H atoms are more visible now.

Fig. S12.1. Geometries of all computed model with two copper atoms deposited on MOF-808. See table S11.1 for their energetics.

Fig. S12.2. Optimized geometries of NO_2 interacting with the Cu-MOF-808, MOF-808 and Cu-HKUST-1 frameworks.

5. In the SI section “Results”, page 18, the authors say “Subsequently, based on the observation for the deposition of one copper atom, several possible conformations have been considered for the deposition of two copper atoms.” The term “conformation” in chemistry is related to the isomerism due to rotation of functional groups or part of the molecule around single bond. Is the term “conformation” used correctly here? I would rephrase the sentence using “configuration” instead of “conformation”, because on fig S11.1 are shown structures with different mutual arrangements of the two Cu ions.

We thank the referee for identifying this mistake in the manuscript. We have now replaced the word “conformation” by “configuration” both in the text and SI. The specific changes made in the main text are indicated below.

Page 6 – “Density Functional Theory (DFT) calculations were performed in order to elucidate the precise conformations of Cu-MOF-808.” This sentence has been replaced by: “Density Functional Theory (DFT) calculations were performed in order to elucidate the precise configurations of Cu-MOF-808.”

Page 6 – “Here, the three most stable conformations are presented, for which the most negative formation energies have been calculated.” This sentence has been replaced by: “Here, the three most stable configurations are presented, for which the most negative formation energies have been calculated.”

Page 7 – “Among them, the most stable conformation is with a μ_2 -OH bridge (model C-2Cu), and is ~35 kJ/mol (14 kJ/mol) higher than model A-2Cu according to the r2-SCAN-3c (M06L),”. This sentence has been replaced by “Among them, the most stable configuration is with a μ_2 -OH bridge (model C-2Cu), and is ~35 kJ/mol (14 kJ/mol) higher than model A-2Cu according to the r2-SCAN-3c (M06L),”.

Page 8 – Caption of Figure 5 “... The three most stable conformations are presented here..” has been replaced by “... The three most stable configurations are presented here...”

Page 11 – “Several initial conformations have been considered and the interaction energies of NO₂ with MOF-808 are computed between ~-22 and ~-29 kJ/mol.” This sentence has been replaced by: “Several initial configurations have been considered and the interaction energies of NO₂ with MOF-808 are computed between ~-22 and ~-29 kJ/mol.”

Page 16 – “Density Functional Theory (DFT) calculations were performed in order to elucidate the possible conformations of Cu-MOF-808.” This sentence has been replaced by: “Density Functional Theory (DFT) calculations were performed in order to elucidate the possible configurations of Cu-MOF-808.”

Page 17 – “As a final step, after obtaining the most stable conformations for the deposited Cu(II) atoms, the interactions with NO₂ have been computed.” This sentence has been replaced by: “As a final step, after obtaining the most stable configurations for the deposited Cu(II) atoms, the interactions with NO₂ have been computed.”

Page 17 – “Several initial conformations have been considered, their geometries have been optimized,”. This sentence has been replaced by: “Several initial configurations have been considered, their geometries have been optimized,”

Summary

In conclusion, the manuscript presents successfully incorporated synergistic binding sites in MOF-808, resulting in excellent optical sensing properties towards NO₂ at remarkably low concentrations . The combination of experimental techniques with computational modeling is used to get insight in to the atomistic structure and host-guest interactions in the system under study. This work opens new prospects in the field of optical sensing of toxic gases using MOF materials by tuning the host-guest interactions through post-synthetic chemical modifications. If the authors improve the manuscript and the SI answering the questions raised above, I would suggest the manuscript as suitable for publication in Nature communication journal.

Summary

This manuscript discusses the synthesis and characterization of a Cu-MOF-808 structure, based on post-synthetic processing of the MOF. The MOF-808 structure remains even after incorporation of the Cu, and appears to be stable, with an even distribution of Cu. The composition of the metal cluster varies with the type of Cu-precursor used, with a maximum Cu concentration of 3.3 per Zr-cluster. Limited changes in the structure were noted, along with increased adsorption of NO₂ accompanied by changes in luminescence.

The authors have not made a compelling case that the MOF structures are demonstrating a novel or unique capability relative to other MOF structures with known NO₂ sensing capabilities, even at low (ppm/ppb) concentration or low temperature, see some examples below. That said, the incorporation of Cu into MOF-808 is unique but its particular application in this area is less clear.

- <https://pubs.acs.org/doi/full/10.1021/acsami.0c10420>
- <https://pubs.acs.org/doi/full/10.1021/acsami.0c00803>
- <https://onlinelibrary.wiley.com/doi/full/10.1002/adfm.202006598>
- <https://www.sciencedirect.com/science/article/pii/S0925400520314003>

Although the highlighted articles are remarkable in terms of sensing of NO₂, we believe that these materials cannot be compared with the optical properties of Cu-MOF-808 reported in this work. The first work is based on the product obtained after calcination of a MOF precursor – the ultimate material is a metal-oxide such as In-doped ZnO with semiconductor behaviour with electrical response for the detection of NO₂ electrical sensing (*ACS Appl. Mater. Interfaces* **2020**, *12*, 37489). The other mentioned works are based on MOFs with chemoresistive rather than optical transduction. The reported MOFs are a Cu-MOF with an unknown structure (*Sensor. Actuat. B-Chem.*, **2021**, *329*, 129053), a MOF-composite based on PDVT-10 and [Ni(TPyP)-(TiF6)] (*ACS Appl. Mater. Interfaces*, **2020**, *12*, 18748), and a Ni-MOF-74 (*Adv. Funct. Mater.* **2020**, *30*, 2006598). Furthermore, it is worth highlighting that our new experiments demonstrate LOD values for Cu-MOF-808 of 16 ppb towards the optical sensing of NO₂, while the two other MOF materials reported in these works exhibit responses of 0.14 and 0.5 ppm for Cu-MOF and Ni-MOF-74, respectively.

Simulations are used to predict the stable locations of the Cu in the metal clusters of the Cu-MOF-808 structure based on a simplified gas-phase model of the system composed of two metal clusters linked by two ligands. The authors have used previously used similar clusters for evaluation of these MOF systems. The methodology used to evaluate the formation enthalpy is typically used in literature, where the energy balance of the reaction is used to predict the thermodynamic stability of the product. Overall, the conclusions are supported by the presented evidence and the methodology is sound with sufficient detail to replicate the results. However, there are several comments that can be addressed:

Comments

1. The replacement of the Zr by the Cu justifies the evaluation of the minimized cluster model. Either 1 or 2 Cu atoms in the cluster model are evaluated for their binding. In the experimental portion, it was proposed that there is quite a bit more Cu metal clusters in the model. Why were high numbers of Cu clusters included in the model? That would be more representative of the structure itself and any distortion that may be arising from this high concentration of Cu per cluster.

We thank the referee to raise these concerns and give us the opportunity to clarify this point. In this work, we propose that the copper sites are attached to the Zr_6O_8 clusters within MOF-808 through binding to the $-OH/H_2O$ groups – it is not a doping scenario where zirconium is replaced by copper. The experimental evidence coming from synchrotron characterization together with the computational modelling, suggest that the copper single sites are located between Zr_6O_8 nodes and along the tetrahedral units. Due to the symmetry of the framework, it is worth clarifying that every Zr_6O_8 in MOF-808 is surrounded by 6 Zr_6O_8 connected through 6 edges (Figure 1b) – that means that our structural model can explain up to a total loading of 6 Cu atoms per Zr_6O_8 node. However, full functionalization in MOFs is unlikely to be achieved post-synthetically – it is indeed difficult to imagine that all the potential binding sites in the MOF-808 are occupied by copper. Chemical analyses of Cu-MOF-808 show a ratio of 3.3 Cu to Zr_6O_8 , which would suggest that around *ca.* 50 % of the edges will be decorated with copper sites.

Then, to identify the size of the copper sites, we carried out PDF simulations by considering copper clusters

of different sizes. Similar approach has been proven before to be useful to identify the size of nickel- (*J. Am. Chem. Soc.* **2017**, 139, 10410) and iron-oxo (*Chem. Commun.* **2020**, 56, 15615) clusters deposited in Zr-MOFs. By increasing the cluster size, the ratio of $Cu\cdots Cu$ to $Cu\cdots O$ distances increases rapidly. Our PDF simulations show that the presence of copper dimers or trimers would give rise to signals of increasing intensity linked to $Cu\cdots Cu$ distances (Figure 2). Contrary, the experimental differential PDF data demonstrated the absence of $Cu\cdots Cu$ correlations in Cu-MOF-808, and therefore the occurrence of copper single sites (Figure 4, main text). This was an important piece of information to build up the structural models reported in this work.

Figure 1. A) Structural model (B-2Cu) showing the deposition of copper single sites along the edges bridging two Zr_6O_8 clusters. **B)** Representation of two corner-sharing tetrahedral cavities within MOF-808, showing that every Zr_6O_8 cluster is surrounded by six more clusters.

Figure 2. Simulated PDF profiles of different copper clusters of different sizes (i.e. 1, 2 and 3 Cu centers), showing all the Cu-O, Cu...O and Cu...Cu distances

2. The structural analysis seems to indicate that in some cases there may be sufficient deposited Cu in the pore structure to entirely fill some of the smaller pore spaces (seen by the decrease in surface area for the Cu-MOF-808 data). Since the NO₂ looks to be binding primarily with the Cu what does the MOF-808 structure provide for NO₂ binding/sensing? Is it serving primarily as a scaffold? Would this process exist in other MOF structures?

Our results demonstrate that the copper single sites are bridging the Zr₆O₈ nodes, rather than filling the small pores. The deposition of the copper single sites along the edges of the small pores in MOF-808 would explain both the decrease in micropore volume and the surface area, as observed by nitrogen isotherms (Table S6.1). Moreover, if the pores were filled by copper oxide nanoparticles, the PDF would give rise to signals of increasing intensity linked to Cu...Cu distances (shown above in Figure 2) due to presence of copper dimers, trimers or larger clusters. Contrarily, the experimental differential PDF data demonstrated the absence of Cu...Cu correlations in Cu-MOF-808, and therefore the occurrence of copper single sites. In summary, the results from N₂ adsorption and simulated PDF profiles (see above Figure 2) fit very well with the proposed structural model.

Our computational studies based on Interaction Region Indicator (IRI) analysis suggest that the adsorption mechanism of NO₂ in the Cu-MOF-808 can be explained as a combination of weak dispersive- (with the Zr₆O₈ cluster) and stronger metal-bonding (with the copper sites) interactions. In other words, the MOF-808 scaffold plays a synergistic role towards the adsorption of NO₂. This is an unprecedented mechanism seen in MOFs, arising from the unique atomic structure of Cu-MOF-808 composed of heterobimetallic Zr(IV)-Cu(IV)-oxo tetrahedral

units. We believe that this finding may open new synthetic strategies in the field of MOF for the development of more efficient gas sensors, beyond NO₂.

3. On a more methodological note, it is not clear why Cu(OH)₂(H₂O)₂ is selected as a precursor. If the intention was to use hydrated Cu²⁺ ions then a five-fold coordination is more favored (<https://www.science.org/doi/10.1126/science.291.5505.856>).

The model of the copper precursor was chosen so that we could construct a hypothetical reaction and calculate the reaction energy, where protons are abstracted from the ZrO₂ nodes and Cu(OH)_x(H₂O)_y species are deposited. It is not possible to compare directly the energies of all different Cu_{1,2}O_x(OH)_y(H₂O)_z clusters presented in Table S12.1, because they possess different stoichiometries. Models A to F have same stoichiometries, therefore a direct conclusion about the stability can only be done by comparing the energies of models A to F. A comparison among all models can be made, only if a hypothetical reaction is taken into account. For that reason, a Cu(OH)₂(H₂O)₂ was considered as precursor. The relative order of the formation energies of the clusters will not change, if a different precursor is considered. For example, if we consider Cu(CH₃COO)₂(H₂O)₂ or Cu²⁺(OH)₅ precursors, the formation energies of all models are shown in the following Table 3. The relative formation energies will not change and do not depend on the choice of the copper precursor. Similar approaches have been used in our previous works: i) the palladium precursor was modelled as PdCl₂(CH₃CN)₂ in *Angew. Chem. Int. Edt.* **2020**, 59, 13013-13020, ii) the iron precursor was modelled as FeCl₃ in *Chem. Commun.* **2020**, 56, 15615-15618.

Table 3. Formation energies ΔE^{form} and relative formation energies $\Delta\Delta E^{form}$ (in kJ mol⁻¹) for the deposition of two copper-hydroxide species on the MOF-808 nodes starting from three different copper precursors: Cu(OH)₂(H₂O)₂, Cu(CH₃COO)₂(H₂O)₂, and [Cu(H₂O)₅]²⁺.

Model	Precursor					
	Cu(OH) ₂ (H ₂ O) ₂		Cu(CH ₃ COO) ₂ (H ₂ O) ₂		[Cu(H ₂ O) ₅] ²⁺	
	ΔE^{form} (kJ/mol)	$\Delta\Delta E^{form}$ (kJ/mol)	ΔE^{form} (kJ/mol)	$\Delta\Delta E^{form}$ (kJ/mol)	ΔE^{form} (kJ/mol)	$\Delta\Delta E^{form}$ (kJ/mol)
A-2Cu	-248.8	0	3.2	0	157.6	0.0
B-2Cu	-208.3	40.5	43.7	40.5	198.2	40.6
C-2Cu	-213.9	34.9	38.1	34.9	192.5	34.9
D-2Cu	-185.0	63.8	67.0	63.8	221.4	63.8
E-2Cu	-168.4	80.4	83.6	80.4	238.0	80.4
F-2Cu	-140.7	108.1	111.3	108.1	265.7	108.1
G-2Cu	-21.3	227.5	230.7	227.5	385.2	227.6
H-2Cu	12.3	261.1	264.3	261.1	418.8	261.2
I-2Cu	15.1	263.9	267.1	263.9	421.6	264.0
J-2Cu	151.6	400.4	403.6	400.4	558.1	400.5

4. There was a missed opportunity to used excited state calculations to directly probe the impact of the NO₂ adsorption on the luminesce properties of the Cu-Zr metal clusters. In particular, the charge transfer to the LUMO orbital is the main comment on the source of the change in the optical properties. This appears to be more of a side comment, and a more thorough investigation of the source of the optical changes would greatly strengthen the impact of the manuscript.

We thank the reviewer for suggesting these calculations. Indeed, the excited state calculations could provide important information to demonstrate the effect of the NO₂ interactions on the absorption spectra of Cu-MOF-808. This was also shown in a previous publication (*J. Phys. Chem. Lett.* 2020, 11, 9, 3362–3368) of one of the authors of the current work, where the excited states of the organic ligand of a MOF were computed upon interaction with a NO₂ molecule. However, the usage of TD-DFT is not adequate in predicting accurately these excitations, and we would have to use the very time-consuming and resources-demanding CASSCF/CASPT2 calculations. Such calculations are restricted to small-sized molecular systems and can not be performed to a system of ~160 atoms of the current work.

This manuscript by Mavrandonakis, Platero-Prats and co-workers report the incorporation of Cu in the Zr-based MOF-808 by post-functionalization, its characterization by various techniques including Pair Distribution Function and X-ray absorption spectroscopy and its luminescent properties that can be exploited to obtain an optical sensor for the detection of low concentrations of NO₂. I do not recommend the publication of this paper in Nature Communications for the following reasons:

1) The first reason is the lack of originality: this work follows two previous reports from the same group that are closely related, the first one on Fe-oxo clusters stabilized on MOF-808 (ref. 15) and the second one on MOF-808 decorated with single copper sites (*ACS Applied Mater. Interfaces* **2022**, *14*, 27040, not cited). The originality of the paper could lie in the performances of the material as sensor. This property has not been previously investigated in the two previous reports. However, the significance of this property must be shown. Could the authors compare the performance of their material to other materials reported in the literature? Furthermore, what is the selectivity of this material for NO₂ compared to other gases?

We thank the referee for giving us the opportunity to clarify this point. Although we have previously reported on the stabilization of metal-oxo sites (*Chem. Commun.* **2020**, *56*, 15615–15618) and metal complexes (*ACS Appl. Mater. Interfaces* **2022**, *14*, 27040–27047) on MOF-808 for catalytic transformations, in this work we demonstrate the deposition of synergistic binding meta sites in this material with optical response for the detection of NO₂ at sub-ppm level. According to our LOD results, the limit of detection (LOD) for Cu-MOF-808 and MOF-808 was estimated to be 16 ppb and 101 ppb, respectively. We provide not only insights on the local structure of this copper sites within MOF-808 but also mechanistic insights based on computational modelling. The synergistic effect between the copper and the Zr6O8 clusters results in an unprecedented mechanism for the binding of NO₂ into the MOF, which we consider the main novelty of this work.

We have now included Table S13.1 in the SI to show the performance of Cu-MOF-808 compared to other materials reported in the literature. While there are reported works that show the performance of some MOFs to detect NO₂ gas at very low concentrations (at ppb level), their sensing mechanism are not similar to the one reported in this work. As shown in the following table, the most common methods to detect NO₂ are based on chemoresistivity, and organic field-effect transistors (OFETs) that measure electrical resistance and electrical conductivity. Optical sensing is a fast and simple way to detect NO₂ gas compared to other methods. The development of novel materials with unique optical properties towards the detection of NO₂ at sub-ppm level can afford the fabrication of cost-effective and portable devices.

Table S13.1. Comparison of different materials employed as NO₂ sensors reported in the literature.

Material	Sensor type	T (°C)	Concentration, LOD	Response	Ref.
In/ZnO-10	Chemoresistive	300	0.20 ppb	Electrical resistance	[1]
[Ni(TPyP)-(TiF ₆)]@PDVT-10	OFET/ Chemoresistive	25	8.25 ppb	Electrical resistance	[2]
Co ₃ O ₄ /Biomass carbon	Chemoresistive	25	10 ppb	Electrical conductivity	[3]
In ₂ O ₃ /ZIF-8	Chemoresistive	140	10 ppb	Electrical resistance	[4]
Cu ₃ (HHTP) ₂ /Fe ₂ O ₃	Chemoresistive	20	11 ppb	Electrical resistance	[5]
CuO tube-like nanofibers	Chemoresistive	25	12.2 ppb	Electrical resistance	[6]
PDVT-10	OFET/ Chemoresistive	25	25 ppb	Electrical resistance	[7]
Ti-FIR-120	Chemoresistive	25	40 ppb	Electrical resistance	[8]
Pt@Cu ₃ (HHTP) ₂ thin-film	Chemoresistive	25	0.1 ppm	Electrical resistance	[9]
Cu ₃ (HHTP) ₂ thin-film	Chemoresistive	25	0.1 ppm	Electrical resistance	[9]
Cu-MOF250 (unknown structure)	Chemoresistive	40	0.14 ppm	Electrical resistance	[10]
Ag-3 doped-WO ₃	Chemoresistive	30	196.8 ppb	Electrical resistance	[11]
Ni-MOF-74	Capacitive	50	0.5 ppm	Electrical resistance	[12]
Pd@Cu ₃ (HHTP) ₂ thin-film	Chemoresistive	25	1 ppm	Electrical resistance	[13]
Pt@Cu ₃ (HHTP) ₂ thin-film	Chemoresistive	25	1 ppm	Electrical resistance	[13]
NiFe ₂ O ₄ nanofibers	Chemoresistive	350	10 ppm	Electrical resistance	[14]
((H ₃ O) ₄ [Co ₂ (L)(DMF)(H ₂ O) ₄] ₂ DMF·3H ₂ O)	Chemoresistive	25	-	Electrical conductivity	[15]
ZIF-8	Chemoresistive	350	-	Electrical resistance	[16]
Porphyrin LB Films	Optical	25	0.46 ppm	Red-shift of UV-Vis spectrum	[17]
TCPC@MOF@PDMS	Optical	25	> 0.5 ppm	PL quenching	[18]
Porphyrin-film@SiO ₂	Optical	25	1.0 ppm	UV-Vis peaks shift/Intensity change and turn-on absorption	[19]
{[Tb ₂ (NBDC) ₃ (DMF) ₄]2DMF}	Optical	25	1.8 ppm	PL quenching	[20]
{[Eu ₂ (NBDC) ₃ (DMF) ₄]2DMF}	Optical	25	2.2 ppm	PL enhancement	[20]
Tb(BTC)@PDMS	Optical	25	4 ppm	PL quenching	[21]
Calixarene-based Zr-MOF film	Optical/ Chemoresistive	25	5 ppm	Colorimetric/ Turn-on UV-vis absorption/photo-current change	[22]
Zn-ZJU-66 film	Optical	25	-	PL quenching	[23]
Y-DOBDC	Optical	25	-	Colorimetric/ PL quenching	[24]
((H ₃ O) ₄ [Co ₂ (L)(DMF)(H ₂ O) ₄] ₂ DMF·3H ₂ O)	Optical/ Chemoresistive	25	-	Colorimetric/ Conductivity	[25]

MOF-808	Optical	25	101 ppb	PL quenching	In this work
Cu- MOF-808	Optical	25	16 ppb	PL quenching	In this work

References:

- Li, Z. Zhang, Y. Zhang, H. Jiang, Y. and Yi, J. Superior NO₂ sensing of MOF-derived indium-doped ZnO porous hollow cages. *ACS Appl. Mater. Interfaces*, **12**, 37489-37498 (2020).
- Yuvaraja, S. Surya, S.G. Chernikova, V. Vijjapu, M.T. Shekhah, O. Bhatt, P.M. Chandra, S. Eddaoudi, M. and Salama, K.N. Realization of an ultrasensitive and highly selective OFET NO₂ sensor: the synergistic combination of PDVT-10 polymer and porphyrin-MOF. *ACS Appl. Mater. Interfaces*, **12**, 18748-18760 (2020).
- Chen, J, Lv, H. Bai, X. Liu, Z. He, L. Wang, J. Zhang, Y. Sun, B. Kan, K. and Shi, K. Synthesis of hierarchically porous Co₃O₄/Biomass carbon composites derived from MOFs and their highly NO₂ gas sensing performance. *Microporous Mesoporous Mater*, **321**, 111108 (2021).
- Liu, Y. Wang, R. Zhang, T. Liu, S. and Fei, T. Zeolitic imidazolate framework-8 (ZIF-8)-coated In₂O₃ nanofibers as an efficient sensing material for ppb-level NO₂ detection. *J. Colloid Interface Sci*, **541**, 249-257 (2019).
- Jo, Y.M. Lim, K. Yoon, J.W. Jo, Y.K. Moon, Y.K. Jang, H.W. and Lee, J.H. Visible-light-activated Type II heterojunction in Cu₃(hexahydroxytriphenylene)₂/Fe₂O₃ hybrids for reversible NO₂ sensing: Critical role of π - π^* transition. *ACS Cent. Sci.*, **7**, 1176-1182 (2021).
- Liu, J. Wang, W. Li, G. Bian, X. Liu, Y. Zhang, J. Gao, J. Wang, C. Zhu, B. and Lu, H. Metal-organic framework-derived CuO tube-like nanofibers with high surface area and abundant porosities for enhanced room-temperature NO₂ sensing properties. *J. Alloy. Comp.* **934**, 167950 (2023).
- Yuvaraja, S. Surya, S.G. Vijjapu, M.T. Chernikova, V. Shekhah, O. Eddaoudi, M. and Salama, K.N. Fully Integrated Organic Field-Effect Transistor Platform to Detect and to Quantify NO₂ Gas. *Phys. Status Solidi RRL*, **14**, 2000086 (2020).
- Li, H.Z. Pan, Y. Li, Q. Lin, Q. Lin, D. Wang, F. Xu, G. and Zhang, J. Rationally designed titanium-based metal-organic frameworks for visible-light activated chemiresistive sensing. *J. Mater. Chem. A*, **11**, 965 (2023).
- Kim, J.O. Koo, W.T. Kim, H. Park, C. Lee, T. Hutomo, C.A. Choi, S.Q. Kim, D.S. Kim, I.D. and Park, S. Large-area synthesis of nanoscopic catalyst-decorated conductive MOF film using microfluidic-based solution shearing. *Nat. Commun.* **12**, 4294 (2021).
- Arul, C. Moulae, K. Donato, N. Iannazzo, D. Lavanya, N. Neri, G. and Sekar, C. Temperature modulated Cu-MOF based gas sensor with dual selectivity to acetone and NO₂ at low operating temperatures. *Sensor. Actuat. B-Chem.* **329**, 129053 (2021).
- Mathankumar, G. Harish, S. Mohan, M.K. Bharathi, P. Kannan, S.K. Archana, J. and Navaneethan, M. Enhanced selectivity and ultra-fast detection of NO₂ gas sensor via Ag modified WO₃ nanostructures for gas sensing applications. *Sensor. Actuat. B-Chem.* 133374 (2023).

12. Small, L.J. Henkelis, S.E. Rademacher, D.X. Schindelholz, M.E. Krumhansl, J.L. Vogel, D.J. and Nenoff, T.M. Near-zero power MOF-based sensors for NO₂ detection. *Adv. Funct. Mater.* **30**, 2006598 (2020).
13. Koo, W.T. Kim, S.J. Jang, J.S. Kim, D.H. and Kim, I.D. Catalytic metal nanoparticles embedded in conductive metal–organic frameworks for chemiresistors: highly active and conductive porous materials. *Adv. Sci.* **6**, 1900250 (2019).
14. Van Hoang, N. Hiep, N.T. Hung, N.M. Nguyen, C.V. Hung, P.T. Hoat, P.D. and Heo, Y.W. Optimization of synthesis conditions and sensing performance of electrospun NiFe₂O₄ nanofibers for H₂S and NO₂ detection. *J. Alloy. Comp.* **936**, 168276, (2023).
15. Ma, Y.X. Gao, B. Li, Y. Wei, W. Zhao, Y. and Ma, J.F. Macrocyclic-Based Metal–Organic Frameworks with NO₂-Driven On/Off Switch of Conductivity. *ACS Appl. Mater. Interfaces*, **13**, 27066–27073 (2021).
16. Zhan, M. Hussain, S. AlGarni, T.S. Shah, S., Liu, J. Zhang, X. Ahmad, A. Javed, M.S. Qiao, G. and Liu, G. Facet controlled polyhedral ZIF-8 MOF nanostructures for excellent NO₂ gas-sensing applications. *Mater. Res. Bull.* **136**, 111133 (2021).
17. Pedrosa, J.M. Dooling, C.M. Richardson, T.H. Hyde, R.K. Hunter, C.A. Martin, M.T. and Camacho, L. Characterization and fast optical response to NO₂ of porphyrin LB films. *Mater. Sci. Eng. C*, **22**, 433–438 (2002).
18. Queirós, C. Moscoso, F.G. Almeida, J. Silva, A.M. Sousaraei, A. Cabanillas-González, J. Ribeiro Carrott, M. Lopes-Costa, T. Pedrosa, J.M. and Cunha-Silva, L. MOF-Based Materials with Sensing Potential: Pyrrolidine-Fused Chlorin at UiO-66 (Hf) for Enhanced NO₂ Detection. *Chemosensors* **10**, 511 (2022).
19. Gulino, A. Mineo, P. Scamporrino, E. Vitalini, D. and Fragalà, I. Molecularly engineered silica surfaces with an assembled porphyrin monolayer as optical NO₂ molecular recognizers. *Chem. Mater.* **16**, 1838–1840 (2004).
20. Gamonal, A. Sun, C., Mariano, A.L. Fernandez-Bartolome, E. Guerrero-SanVicente, E., Vlaisavljevich, B. Castells-Gil, J. Marti-Gastaldo, C. Poloni, R. Wannemacher, R. and Cabanillas-Gonzalez, J. Divergent adsorption-dependent luminescence of amino-functionalized lanthanide metal–organic frameworks for highly sensitive NO₂ sensors. *J. Phys. Chem. Lett.* **11**, 3362–3368 (2020).
21. Moscoso, F.G. Almeida, J. Sousaraei, A. Lopes-Costa, T. Silva, A.M. Cabanillas-Gonzalez, J., Cunha-Silva, L. and Pedrosa, J.M. Luminescent MOF crystals embedded in PMMA/PDMS transparent films as effective NO₂ gas sensors. *Mol. Syst. Des. Eng.* **5**, 1048-1056 (2020).
22. Schulz, M. Gehl, A. Schlenkrich, J. Schulze, H.A. Zimmermann, S. and Schaate, A. A Calixarene-Based Metal–Organic Framework for Highly Selective NO₂ Detection. *angew. Chem. Int. Ed.* **57**, 12961–12965 (2018).
23. Zhang, J. Hu, E., Liu, F. Li, H. and Xia, T. Growth of robust metal–organic framework films by spontaneous oxidation of a metal substrate for NO₂ sensing. *Mater. Chem. Front.* **5**, 6476 (2021).
24. Sava Gallis, D.F. Vogel, D.J. Vincent, G.A. Rimsza, J.M. and Nenoff, T.M. NO_x adsorption and optical detection in rare earth metal–organic frameworks. *ACS Appl. Mater. Interfaces*, **11**, 43270–43277 (2019).

25. Ma, Y.X. Gao, B. Li, Y. Wei, W. Zhao, Y. and Ma, J.F. Macrocycle-Based Metal–Organic Frameworks with NO₂-Driven On/Off Switch of Conductivity. *ACS Appl. Mater. Interfaces*, **13**, 27066–27073 (2021).

To assess the sensitivity of Cu-MOF-808 to other gases, we have carried out new experiments to determine the limit of detection of both MOF-808 and Cu-MOF-808 to NO₂ and other relevant gases found in air such as CO₂ and O₂. These gases may be able to potentially bind to the copper sites and poison the optical response of this material. Interestingly, our results indicate significantly higher sensitivity of Cu-MOF-808 towards NO₂ (0.016 ppm) than CO₂ (2837 ppm) and O₂ (476 ppm). In the case of MOF-808, we have observed a different behaviour to sense CO₂, followed by a subtle PL enhancement which makes difficult to determine the LOD.

The LOD values were calculated based on the standard deviation (σ) obtained from measuring the PL spectra of the materials under N₂ (0 ppm NO₂) at fourteen separate spectra. Then, after linear fitting of the materials under exposure to different gases, the slopes were incorporated in the following formula:

$$\text{LOD} = 3.3 \sigma / \text{slope}$$

Table 4. LOD values of MOF-808 and Cu-MOF-808 towards NO₂, CO₂ and O₂.

Material	LOD (NO ₂)	LOD (CO ₂)	LOD (O ₂)
MOF-808	0.101 ppm	-	588 ppm
Cu-MOF-808	0.016 ppm	2837 ppm	476 ppm

We have now added the following sentence in the main text:

Page 9 – “...longer respond of NO₂ gas. The materials were then exposed to different concentrations of NO₂ to assess the limit of detection (LOD), resulting in values of 101 and 16 ppb for MOF-808 and Cu-MOF-808, respectively. Furthermore, the sensitivity of the Cu-MOF-808 to other common gases present in air potentially able to bind to the copper sites were explored, such as CO₂ and O₂. The results confirmed a significantly higher sensitivity of Cu-MOF-808 to NO₂ compared to CO₂ and O₂ (see Figure S12.6). These results demonstrate the best optical response reported so far for a MOF material towards the sensing of NO₂”.

We have also added two new figures in the SI (Figure S13.6-7) to show the sensitivity of MOF-808 and Cu-MOF-808 to NO₂, CO₂ and O₂.

Fig. S13.6. Linear fitting the PL quenching of MOF-808 and Cu-MOF-808 exposed to different concentrations of NO₂ (0 – 10 ppm).

Fig. S13.7. PL quenching of MOF-808 and Cu-MOF-808 exposed to different concentrations of NO₂, CO₂ and O₂ after activating with N₂ (λ_{exc} : 355 nm).

As also mentioned above in our reply to the Comment 1 of Reviewer 1, we have explored computationally the strength of the binding between competing molecules (such as H₂O, NO, CO and CO₂) with the Cu-MOF-808. The relative strength of the interactions between the four competing molecules (NO₂, NO, CO, CO₂ and H₂O) and the copper site is used as a qualitative descriptor to predict, which of the above molecules will be preferentially adsorbed. In conclusion, the calculations suggest that NO, CO and CO₂ are adsorbed less strongly than the NO₂, and that at humid conditions one of the two copper sites will be able to bind NO₂. For a more detailed answer, we ask the reviewer to read our reply to the Comment 1 of Reviewer 1 in the pages 4 and 5 of this response letter.

2) My second main concern is about the chemical formula proposed in the experimental section: $[\text{Zr}_6\text{Cu}_{3.3}\text{O}_8(\text{C}_9\text{H}_3\text{O}_6)_2(\text{H}_2\text{O})_{25}]$. This formula obviously does not respect the electroneutrality. There is clearly a lack of negative charges. C, H, N analysis must be given.

We thank the referee for identifying this mistake in the manuscript. We have now performed elemental analysis and liquid $^1\text{H-NMR}$ on the digested MOF samples, before and after modification with copper. These experimental data suggest the following chemical formula:

Elemental analyses: *Calcd.:* C 13.85%, H 4.51 %, N 0.38%; *Found:* C 13.23%, H 4.21%, N 1.07%.
 $^1\text{H-NMR}$: 8.63 (s, 6H, 2 × BTCs), 8.11 (s, 1.9H, 1.7 × HCOO-), 2.87 (s, 1H, 0.3 × DMF), 2.72 (s, 1H, 0.3 × DMF).

Chemical analysis by ICP showed a Cu/Zr₆ ratio of 3.3.

Elemental analyses: *Calcd.:* C 17.97%, H 2.23%, N 0.32%; *Found:* C 19.50%, H 2.61%, N 0.20%.
 $^1\text{H-NMR}$: 8.62 (s, 6H, 2 × BTCs), 2.85 (s, 0.5H, 0.17 × DMF), 2.73 (s, 0.5H, 0.17 × DMF), 1.89 (s, 9H, 3 × acetates).

We have included this information in the Supporting Information (Section S2).

3) Finally the paper is confusing about the position of the Cu ions in the MOF and the agreement of DFT, EXAFS and DFT calculations about the Cu···M interactions. For example it is written page 6 line 120 that “According to both the EXAFS data and the differences of X-ray scattering power of Cu and Zr, the most plausible assignment is Cu···Zr”. However the existence of Cu···Zr interactions is not mentioned in the EXAFS section. It is also written page 7 line 45 that “The characteristics, i.e the distorted planar coordination environment of the copper atom and the Cu-Cu and Cu···Zr distances, for models B and C indicate very good agreement with the experimental PDF/EXAFs data.” However, PDF and EXAFS data do not show the existence of Cu-Cu interactions.

We thank the referee to give us the opportunity to clarify these concerns about the structural analysis of Cu-MOF-808. EXAFS experiments have been key to study the coordination environment of copper within MOF-808. The EXAFS data of Cu-MOF-808 is dominated by a signal at $\sim 1.6 \text{ \AA}$ together with the lack of significant contributions beyond $\sim 2.0 \text{ \AA}$, which is characteristic of the presence of isolated copper cations. The presence of copper-oxo dimers or trimers would imply the presence of intense EXAFS peaks between $\sim 2.0\text{-}2.5 \text{ \AA}$. These signals are not observed in the EXAFS spectrum of Cu-MOF-808. On the hand, our experience based on similar Zr-MOF materials (*J. Am. Chem. Soc.* **2017**, 139, 30, 10410–10418; *J. Am. Chem. Soc.* **2017**, 139, 42, 15251–15258; *J. Am. Chem. Soc.* **2018**, 140, 45, 15309–15318) has shown that is uncommon to identify Zr···M distances by EXAFS which are located beyond $\sim 3.0 \text{ \AA}$, possibly due to structural disorder. In this context, PDF (which is less sensitive to disorder phenomena) becomes a powerful local probe to identify the occurrence of Zr···M within modified Zr-MOFs, which are indeed indicative of the added-metal sites to be attached to the Zr₆O₈ clusters. This is powerful experimental evidence of the position of the copper ions

within the MOF framework. For Cu-MOF-808, besides the dPDF signal at 2.06 Å linked to Cu-O distances (in agreement with EXAFS), we have identified the presence of an intense peak at 3.34 Å that can be assigned to Zr...Cu distances. This, together with the absence of signals associated with Cu-Cu distances between 2.5-3.0 Å (typical values for copper-oxo clusters), demonstrates the stabilization of copper single sites attached to the Zr₆O₈. This is how far we can go with experimental techniques in terms of elucidating the atomic structure and position of the copper sites within Cu-MOF-808.

By applying computational modelling, we have demonstrated the co-existence of two unique heterobimetallic structures in Cu-MOF-808: B-2Cu (two single sites) and C-2Cu (two single sites linked by one hydroxo ligand). The occurrence of the configuration C-2Cu agrees with the dPDF data, due to the unexpectedly long Cu...Cu distance at 3.35 Å seen in the model (which would overlap with the Cu...Zr distances at 3.30 Å).

To clarify this point, we have made the following changes in the text:

Page 7 - "The characteristics, i.e the distorted planar coordination environment of the copper atom, and the Cu...Cu and Cu...Zr distances, for models B and C indicate very good agreement with the experimental PDF/EXAFs data."

Minor corrections:

1) There should be a more detailed bibliographic paragraph on studies of the introduction of Cu or other metal sites in Zr-based MOFs reported in the literature. For example, the paper published in 2019 in Nature Catalysis of Bing An et al. (doi 10.1038/s41929-019-0308-5) must be cited.

Following the reviewers' comment, we have now expanded the following paragraph in the introduction, to provide a better context of metal decorations in Zr-MOFs.

"Defect engineering in MOFs has acquired remarkable attention in the field of gas adsorption.⁹ Zr-MOFs may have structural defects without compromising their stability.¹⁰ One strategy to incorporate functional defects in robust Zr-MOFs consists of modifying the inorganic Zr₆O₈ clusters with metal cations.¹¹⁻¹⁷ In particular, the incorporation of copper in Zr-MOFs can afford either the stabilization of Cu(II)^{18,19} or Cu(I)¹⁷ as single sites. Within MOF-808, the incorporation of copper can be done ~~Particularly, MOF-808 allows chemical modifications~~ through either insertion of functional groups^{20,21} or metalation of reactive aqua ligands within the Zr₆O₈ clusters.^{22"}

We have also included references [13-19] and [21] in the article.

~~[11] M. J. Kalmutzki, N. Hanikel, O. M. Yaghi, Sci. Adv. **2018**, 4, eaat9180.~~

[11] M. Rimoldi, V. Bernales, J. Borycz, A. Vjunov, L. C. Gallington, A. E. Platero-Prats, I. S. Kim, J. L. Fulton, A. B. F. Martinson, J. A. Lercher, K. W. Chapman, C. J. Cramer, L. Gagliardi, J. T. Hupp, O. K. Farha, Chem. Mater. **2017**, 29, 1058–1068. (Al(III) in NU-1000)

- [12] Noh, H.; Cui, Y.; Peters, A. W.; Pahls, D. R.; Ortuño, M. A.; Vermeulen, N. A.; Cramer, C. J.; Gagliardi, L.; Hupp, J. T.; Farha, O. K. *J. Am. Chem. Soc.* **2016**, *138*, 23. (Mo(VI) in NU-1000)
- [13] Li, Z.; Peters, A. W.; Bernales, V.; Ortuño, M. A.; Schweitzer, N. M.; Destefano, M. R.; Gallington, L. C.; Platero-Prats, A. E.; Chapman, K. W.; Cramer, C. J.; Gagliardi, L.; Hupp, J. T.; Farha, O. K. *ACS Cent. Sci.* **2017**, *3*, 31–38. (Co(II) in NU-1000)
- [14] Otake, K.-I.; Cui, Y.; Buru, C. T.; Li, Z.; Hupp, J. T.; Farha, O. K. *J. Am. Chem. Soc.* **2018**, *140*, 54. (V(V) in NU-1000 and MOF-808)
- [15] Abdel-Mageed, A. M.; Rungtaweevoranit, B.; Parlinska-Wojtan, M.; Pei, X.; Yaghi, O. M.; Jü Rgen Behm, R. *J. Am. Chem. Soc.* **2019**, *141*, 45. (Cu(II) in UiO-66)
- [16] Zheng, J.; Ye, J.; Ortuño, M. A.; Fulton, J. L.; Gutiérrez, O. Y.; Camaioni, D. M.; Motkuri, R. K.; Li, Z.; Webber, T. E.; Mehdi, B. L.; Browning, N. D.; Penn, R Lee; Farha, O. K.; Hupp, J. T.; Truhlar, D. G.; Cramer, C. J.; Lercher, J. A. *J. Am. Chem. Soc.* **2019**, *141*, 56. (Cu(II) in NU-1000)
- [17] An, B.; Li, Z.; Song, Y.; Zhang, J.; Zeng, L.; Wang, C.; Lin, W. *Nat. Catal.* **2019**, *2* (8), 709–717. (Cu(I) in Zr-bpdc)
- [18] J. Jiang, F. Gándara, Y. B. Zhang, K. Na, O. M. Yaghi, W. G. Klemperer, *J. Am. Chem. Soc.* **2014**, *136*, 12844–12847.
- [18] J. Baek, B. Rungtaweevoranit, X. Pei, M. Park, S. C. Fakra, Y. S. Liu, R. Matheu, S. A. Alshimri, S. Alshehri, C. A. Trickett, et al., *J. Am. Chem. Soc.* **2018**, *140*, 18208–18216. (Cu(I) in imidazole-MOF-808)
- [19] Romero-Muñiz, I.; Romero-Muñiz, C.; del Castillo-Velilla, I.; Marini, C.; Calero, S.; Zamora, F.; Platero-Prats, A. E. *ACS Appl. Mater. Interfaces* **2022**, *14*, 27040–27047. (Cu(II) in catechol-MOF-808)
- [20] C. Castillo-Blas, I. Romero-Muñiz, A. Mavrandonakis, L. Simonelli, A. E. Platero-Prats, *Chem. Commun.* **2020**, *56*, 15615–15618. (Fe(III) in MOF-808)

2) The authors indicate page 5 line 108 that the peak centered at ca. 1.6 Å (value without phase correction) corresponds to Cu(II)-O bonds. Could they give the value after correction for non-specialists? Does it correspond to the value found by PDF?

We thank the referee the opportunity to clarify this technical feature. In the case of the EXAFS data, the radial distance (peak position) does not correspond directly to the real space distance (atom-atom distance). This is called the phase correction, and can be quantified by fitting of the experimental data against structural models. Typically, this value is about 0.4 Å (although the precise value can change depending on the experimental conditions of the EXAFS experiments). This is the reason why in the main text we refer to the peak positions instead of the real space distance. Contrary, PDF analyses retrieve directly information in the real space.

We have now fitted the EXAFS and included the phase-corrected value for the Cu-O distances in the main manuscript.

Page 5: “EXAFS data showed a predominant contribution at ca. 1.97 Å peak centered at ca. 1.6 Å (value without phase correction) that corresponds to Cu(II)-O bonds.”

3) Figure 5 is not clear enough, it is not easy to see the coordination sphere around the copper ions. Are there OH groups connected to Zr ions? In this case why do not they appear in the formula (the lack of negative charge would thus be even larger)?

We hope the revised structural representations show in a clear manner the coordination sphere around the copper sites deposited on MOF-808, which can be described as $[\text{Cu}(\text{OH})_3(\text{H}_2\text{O})]^-$ (models B-2Cu and C-2Cu). It is important to clarify that the heterobimetallic Zr(IV)-Cu(II) clusters in MOF-808 are neutral. The decoration of the Zr_6O_8 clusters in MOF-808 **by one Cu(II) site (to simplify)** can be described by the following chemical reactions (guest molecules present in the pores are omitted for clarification purposes):

Pristine MOF-808: $[\text{Zr}_6\text{O}_8\text{H}_4(\text{BTC})_2(\text{COOH})_2(\text{OH})_4(\text{H}_2\text{O})_4]$, where BTC is the benzene-1,3,5-tricarboxylate.

To afford the deposition of Cu(II) sites within the Zr_6O_8 cluster, one H_2O molecule bond to the Zr_6O_8 cluster is deprotonated. This allows the Cu(II) to bind to two OH groups to the Zr(IV) centres. Two more water molecules are included to complete the coordination sphere around Cu(II) to CN=4:

Finally, one water molecule from the coordination sphere of copper is deprotonated to give the final neutral heterobimetallic Zr(IV)-Cu(II) cluster.

We have now edited Fig 5 in the main text as follows.

Figure 5. (A) Formation energies (ΔE^{form}) and relative formation energies ($\Delta\Delta E^{\text{form}}$) in kJ/mol for the deposition of two copper-hydroxide species on the MOF-808 nodes. The three most stable conformers are presented here. The formation energies are calculated by considering a reaction of the MOF-808 with two copper precursor species and the subsequent release of water molecules. The relative formation energies are referred versus the formation energy of the model A-2Cu. (B) Structural and (C) schematic representation of the hetero-bimetallic tetrahedral structural subunit in Cu-MOF-808. Colour scheme: blue = Zr, grey = C, magenta = $\mu_3\text{-O}$, orange = Cu, red = O, green = H.

4) What is the concentration of gas in Figure 6A? This must be indicated in the legend of Figure 6.

We have now edited the caption of Figure 6A, including the concentration of NO₂ used to perform the experiments.

Page 10 - “Figure 6. (A) PL kinetics of Cu-MOF-808 and MOF-808 **under 50 ppm of NO₂**, measuring at 473 nm and 480 nm, respectively, (B) Sensing response of Cu-MOF-808 at 50 (black- squares), 25 (blue- circles), and 10 ppm (red- triangles) concentrations, (C) Sensing performance of Cu-MOF-808 and MOF-808 after 2 h sensing measurements, ($\lambda_{exc}=355$ nm), (D) PXRD analyses of Cu-MOF-808 before and after the NO₂ sensing measurement. (The N₂-NO₂ exposition is shown by the shaded areas).”

5) In Figure 6A, there seems to be a decrease in the response of Cu-MOF-808 with time so that the term “large reversibility through N₂ purging” seems a bit exaggerated.

We have now deleted the word “large” within the sentence.

Page 9 – “According to PL kinetics, Cu-MOF showed significant PL quenching ($\leq 45\%$) in the presence of NO₂ and **large** reversibility through N₂ purging (Figure 6A, blue squares).”

Reviewers' Comments:

Reviewer #1:

Remarks to the Author:

The authors addressed point by point all the raised questions in my report. They performed additional research (experimental and computational) and thus substantially improve the manuscript quality. I suggest the revised version of the manuscript for publication in Nature Communications.

Reviewer #2:

Remarks to the Author:

I would like to thank the authors for their thorough and careful response to the reviewers' points. The additional data and clarification of their results has satisfied all of my concerns regarding this manuscript.

Reviewer #3:

Remarks to the Author:

The authors have addressed all the issues raised by the reviewers and have significantly improved their manuscript. I now recommend its publication in Nature Communications without any further corrections.